# Theoretical Analyses of Hyperparameter Selection in Graph-Based Semi-Supervised Learning

## Abstract

Graph-based semi-supervised learning is a powerful paradigm in machine learning for modeling and exploiting the underlying graph structure that captures the relationship between labeled and unlabeled data. A large number of classical as well as modern deep learning based algorithms have been proposed for this problem, often having tunable hyperparameters. We initiate a formal study of tuning algorithm hyperparameters from parameterized algorithm families for this problem. We obtain novel $O(\log n)$ pseudo-dimension upper bounds for hyperparameter selection in three classical label propagation-based algorithm families, where $n$ is the number of nodes, implying bounds on the amount of data needed for learning provably good parameters. We further provide matching $\Omega(\log n)$ pseudo-dimension lower bounds, thus asymptotically characterizing the learning-theoretic complexity of the parameter tuning problem. We extend our study to selecting architectural hyperparameters in modern graph neural networks. We bound the Rademacher complexity for tuning the self-loop weighting in recently proposed Simplified Graph Convolution (SGC) networks. We further propose a tunable architecture that interpolates graph convolutional neural networks (GCN) and graph attention networks (GAT) in every layer, and provide Rademacher complexity bounds for tuning the interpolation coefficient.

## 1 Introduction

Semi-supervised learning is a powerful paradigm in machine learning which reduces the dependence on expensive and hard-to-obtain labeled data, by using a combination of labeled and unlabeled data. This has become increasingly relevant in the era of large language models, where an extremely large amount of labeled training data is needed. A large number of techniques have been proposed in the literature to exploit the structure of unlabeled data, including popularly used graph-based semi-supervised learning algorithms (Blum & Mitchell, 1998; Zhu et al., 2003; Zhou et al., 2003; Delalleau et al., 2005; Chapelle et al., 2009). More recently, there has been an increasing interest in developing effective neural network architectures for graph-based learning (Kipf & Welling, 2017; Veličković, Petar et al., 2018; Iscen et al., 2019). However, different algorithms, architectures, and values of hyperparameters perform well on different datasets (Dwivedi et al., 2023), and there is no principled way of selecting the best approach for the data at hand. In this work, we initiate the study of theoretically principled techniques for learning hyperparameters from infinitely large semi-supervised learning algorithm families.

In graph-based semi-supervised learning, the graph nodes consist of labeled and unlabeled data points, and the graph edges denote feature similarity between the nodes. There are several classical ways of defining a graph-based regularization objective that depend on the available and predicted labels as well as the graph structure. Optimizing this objective yields the predicted labels and the accuracy of the predictions depends on the chosen objective. The performance of the same objective may vary across datasets. By studying parameterized families of objectives, we can learn to design the objective that works best on a given domain-specific data. Similarly, modern deep learning based techniques often have several candidate architectures and choices for hyperparameters, often manually optimized for each application domain. Recent work has considered the problem of learning the graph hyperparameter used in semi-supervised learning (Balcan & Sharma (2021); Fatemi et al.

(2021)) but leaves the problem of selecting the algorithm hyperparameter wide open. In this paper, we take important initial steps to build the theoretical foundations of algorithm hyperparameter selection in graph-based semi-supervised learning. Note that we focus specifically on algorithm hyperparameters, such as self-loop weights, leaving optimization hyperparameters like learning rates outside the scope of this study.

## 1.1 CONTRIBUTIONS

- We study hyperparameter tuning in three canonical label propagation-based semi-supervised learning algorithms: the local and global consistency (Zhou et al., 2003), the smoothing-based (Delalleau et al., 2005), and a novel normalized adjacency matrix-based algorithm. We prove new $O(\log n)$ pseudo-dimension upper bounds for all three families, where $n$ is the number of graph nodes. Our proofs rely on a unified template based on determinant evaluation and root-counting, which may be of independent interest.

- We provide matching $\Omega(\log n)$ pseudo-dimension lower bounds for all three aforementioned families. Our proof involves novel constructions of a class of partially labeled graphs that exhibit fundamental limitations in tuning the label propagation algorithms.

- Next, we consider the modern graph neural networks (GNNs). We first prove a new Rademacher complexity bound for tuning the weight of self-loops for a popular architecture proposed in Wu et al. (2019), the Simplified Graph Networks (SGC).

- We propose an architecture (GCAN) where a hyperparameter $\eta$ is introduced to interpolate two canonical GNN architectures: graph convolutional neural networks (GCNs) and graph attention neural networks (GATs). We bound the Rademacher complexity of tuning $\eta$.

- We conducted experiments to empirically validate our theoretical findings and demonstrate the effectiveness of our hyperparameter selection framework.

## 1.2 RELATED WORK

**Graph Based Semi-supervised Learning**   Semi-supervised Learning is a popular machine learning paradigm with significant theoretical interest (Zhou et al., 2003; Delalleau et al., 2005; Garg et al., 2020). Classical algorithms focus on label-propagation based techniques, such as Zhou et al. (2003), Zhu et al. (2003), and many more. In recent years, graph neural networks (GNNs) have become increasingly popular in a wide range of application domains (Kipf & Welling, 2017; Veličković, Petar et al., 2018; Iscen et al., 2019). A large number of different architectures have been proposed, including graph convolution networks, graph attention networks, message passing, and so on (Dwivedi et al., 2023). Both label propagation-based algorithms and neural network-based algorithms are useful in real life and perform equally well. For example, although GNN-based algorithms are more predominant in applications, Huang et al. (2020) shows that modifications to label propagation-based algorithms can outperform GNN. For node classification in GNN, many work study generalization guarantees for tuning network weights in GNNs (Oono & Suzuki, 2021; Esser et al., 2021; Tang & Liu, 2023). In contrast, we study the tuning of *hyperparameters*.

**Hyperparameter Selection**   Hyper-parameters, such as the weight for self-loop, play important roles in the performance of both classical methods and GNNs. In general, hyperparameter tuning is performed on a validation dataset, and follows the same procedure: determine which hyperparameters to tune and then search within their domain for the combination of parameter values with best performance Yu & Zhu (2020). Many methods are proposed to efficiently search within the parameter space, such as grid search, random search Bergstra & Bengio (2012), and Bayesian optimization (Mockus (1974); Mockus et al. (1978); Jones et al. (1998)). A few existing works investigate the theoretical aspects of these methods, such as through generalization guarantees and complexities of the algorithms. In particular, Balcan et al. (2024) studies the regularization hyperparameter in Ridge regression, LASSO, and ElasticNet in statistical settings and provides generalization guarantees. For self-supervised learning, Balcan et al. (2019) propose a parameterized algorithm family of clustering algorithms and study the sample and computational complexity of learning the parameters. For semi-supervised learning, a recent line of work (Balcan & Sharma (2021); Sharma & Jones (2023)) considers the problem of learning the best graph hyperparameter from a set of problem instances drawn from a data distribution. However, no existing work theoretically studies the algorithm hyperparameter in semi-supervised learning, or investigates deep semi-supervised learning algorithms

## 2 PRELIMINARIES

**Notations.** Throughout this paper, $f(n) = O(g(n))$ denotes that there exists a constant $c > 0$ such that $|f(n)| \leq c|g(n)|$. $f(n) = \Omega(g(n))$ denotes that there exists a constant $c > 0$ such that $|f(n)| \geq c|g(n)|$. The indicator function is indicated by $\mathbb{I}$, taking values in $\{0, 1\}$. In addition, we define the shorthand $[c] = \{1, 2, \ldots, c\}$. For a matrix $W$, we denote its Frobenius norm by $\|W\|_F$ and spectral norm by $\|W\|$. We also denote the Euclidean norm of a vector $v$ by $\|v\|$.

**Graph-based Semi-supervised Learning.** We are given $n$ data points, where some are labeled, denoted by $L \subseteq [n]$, and the rest are unlabeled. We may also have features associated with each data point, denoted by $z_i \in \mathbb{R}^d$ for $i \in [n]$. We can construct a graph $G$ by placing (possibly weighed) edges $w(u, v)$ between pairs of data points $u, v$. The created graph $G$ is denoted by $G = (V, E)$, where $V$ represents the vertices and $E$ represents the edges. Based on $G$, we can calculate $W \in \mathbb{R}^{n \times n}$ as the adjacency matrix, i.e., $W_{ij} = w(i, j)$. We let $D \in \mathbb{R}^{n \times n}$ be the corresponding degree matrix, so $D = \text{diag}(d_1, \ldots, d_n)$ where $d_i = \sum_{j \in [n]} w(i, j)$.

For a problem instance of $n$ data points, we define input $X$ as $X = (n, \{z_i\}_{i=1}^n, L, G)$, or $X = (n, L, G)$ if no features are available. We denote the label matrix by $Y \in \{0, 1\}^{n \times c}$ where $c$ is the number of classes. Throughout the paper, we assume $c = O_n(1)$, i.e. $c$ is treated as a constant with respect to $n$, which matches most practical scenarios. Here, $Y_{ij} = 1$ if data point $i \in L$ has label $j \in [c]$ and $Y_{ij} = 0$ otherwise. The goal is to predict the labels of the unlabeled data points.

An algorithm $F$ in this setting may be considered as a function that takes in $(X, Y)$ and outputs a predictor $f$ that predicts a label in $[c]$ for each data. We denote $f(z_i)$ as our prediction on the $i$-th data. To evaluate the performance of a predictor $f$, we use 0-1 loss (i.e. the predictive accuracy) defined as $\frac{1}{n} \sum_{i=1}^n \ell_{0-1}(f(z_i), y_i) = \frac{1}{n} \sum_{i=1}^n \mathbb{I}[f(z_i) = y_i]$. In this work, we are interested in the generalizability of an algorithm $F$ on 0-1 loss.

**Hyperparameter Selection.** We consider several *parameterized families* of classification algorithms. Given a family of algorithms $\mathcal{F}_\rho$ parameterized by some parameter $\rho$, and a set of $m$ problem instances $\{(X^{(k)}, Y^{(k)})\}_{k=1}^m$ i.i.d. generated from the data distribution $\mathcal{D}$ of the input space $\mathcal{X}$ and the label space $\mathcal{Y}$, our goal is to select a parameter $\hat{\rho}$ whose corresponding prediction function $f_{\hat{\rho}}$ of algorithm $F_{\hat{\rho}}$ minimizes the prediction error. That is, denote $f_{\hat{\rho}}(z_i^{(k)})$ as the predicted label of data point $z_i^{(k)}$ in the $k$-th problem instance, we want

$$\hat{\rho} = \arg\min_\rho \frac{1}{mn} \sum_{k=1}^m \sum_{i=1}^n \ell_{0-1}(f_\rho(z_i^{(k)}), y_i^{(k)}).$$

Each parameter value $\rho$ defines an algorithm $F_\rho$, mapping a problem instance $(X, Y)$ to a prediction function $f_\rho$, which induces a loss $\frac{1}{n} \sum_{i=1}^n \ell_{0-1}(f_\rho(z_i), y_i)$. We define $H_\rho$ as the function mapping $(X, Y)$ to this loss and $\mathcal{H}\rho = H\rho'_{\rho'}$ as the family of loss functions parameterized by $\rho$.

Note that our problem setting differs from prior theoretical works on graph-based semi-supervised learning. The classical setting considers a single algorithm and learning the model parameter from a single problem instance. We are considering *families of algorithms*, each parameterized by a single hyperparameter, and aiming to learn the best *hyperparameter* across *multiple problem instances*. Our setting combines transductive and inductive aspects: each instance has a fixed graph of size $n$ (transductive), but the graphs themselves are drawn from an unknown meta-distribution (inductive).

**Complexity Measures and Generalization Bounds.** We study the generalization ability of several representative parameterized families of algorithms. That is, we aim to address the question of how many problem instances are required to learn a hyperparameter $\rho$ such that a learning algorithm can perform near-optimally for instances drawn from a fixed problem distribution. Clearly, the more complex the algorithm family, the more number of problem instances are needed.

Specifically, for each algorithm $f_{\hat{\rho}}$ trained given $m$ problem instances, we study the difference in the empirical 0-1 loss and the actual 0-1 on the distribution:

$$\mathbb{E}_{(X,Y)\sim\mathcal{D}}\left[\frac{1}{n}\sum_{i=1}^n \ell_{0-1}(f_{\hat{\rho}}(z_i), y_i)\right] - \min_\rho \mathbb{E}_{(X,Y)\sim\mathcal{D}}\left[\frac{1}{n}\sum_{i=1}^n \ell_{0-1}(f_\rho(z_i), y_i)\right].$$

To quantify this, we consider two learning-theoretic complexity measures for characterizing the learnability of algorithm families: the *pseudo-dimension* and the *Rademacher complexity*.

**Definition 1** (Pseudo-dimension). *Let $\mathcal{H}$ be a set of real-valued functions from input space $\mathcal{X}$. We say that $C = (X^{(1)}, ..., X^{(m)}) \in \mathcal{X}^m$ is **pseudo-shattered** by $\mathcal{H}$ if there exists a vector $r = (r_1, ..., r_m) \in \mathbb{R}^m$ (called "witness") such that for all $b = (b_1, ..., b_m) \in \{\pm 1\}^m$ there exists $H_b \in \mathcal{H}$ such that $sign(H_b(X^{(k)}) - r_k) = b_k$. **Pseudo-dimension** of $\mathcal{H}$, denoted PDIM($\mathcal{H}$), is the cardinality of the largest set pseudo-shattered by $\mathcal{H}$.*

The following theorem bounds generalization error using pseudo-dimension.

**Theorem 2.1.** *(Anthony & Bartlett, 2009) Suppose $\mathcal{H}$ is a class of real-valued functions with range in $[0, 1]$ and finite PDIM($\mathcal{H}$). Then for any $\epsilon > 0$ and $\delta \in (0, 1)$, for any distribution $\mathcal{D}$ and for any set $S = \{X^{(1)}, \dots, X^{(m)}\}$ of $m = O\left(\frac{\text{PDIM}(\mathcal{H})}{\epsilon^2} + \log(\frac{1}{\delta})\right)$ samples from $\mathcal{D}$, with probability at least $1 - \delta$, we have*

$$\left| \frac{1}{m} \sum_{k=1}^{m} H(X^{(k)}) - \mathbb{E}_{X \sim \mathcal{D}}[H(X)] \right| \leq \epsilon, \text{ for all } H \in \mathcal{H}.$$

Therefore, if we can show PDIM($\mathcal{H}_\rho$) is bounded, then using the standard empirical risk minimization argument, Theorem 2.1 implies using $m = O\left(\text{PDIM}(\mathcal{H})/\epsilon^2\right)$ problem instances, the expected error on test instances is upper bounded by $\epsilon$. In Section 3, we will obtain *optimal* pseudo-dimension bounds for three canonical label-propagation algorithm families.

Another classical complexity measure is the Rademacher complexity:

**Definition 2** (Rademacher Complexity). *Given a space $\mathcal{X}$ and a distribution $\mathcal{D}$, let $S = \{X^{(1)}, \dots, X^{(m)}\}$ be a set of examples drawn i.i.d. from $\mathcal{D}$. Let $\mathcal{H}$ be the class of functions $H : \mathcal{X} \to \mathbb{R}$. The **(empirical) Rademacher complexity** of $\mathcal{H}$ is*

$$\hat{R}_m(\mathcal{H}) = \mathbb{E}_\sigma \left[ \sup \left( \frac{1}{m} \sum_{k=1}^{m} \sigma_k H(X^{(k)}) \right) \right],$$

*where each $\sigma_k$ is i.i.d. sampled from $\{-1, 1\}$.*

The following theorem bounds generalization error using Rademacher Complexity.

**Theorem 2.2.** *Mohri et al. (2012) Suppose $\mathcal{H}$ is a class of real-valued functions with range in $[0, 1]$. Then for any $\delta \in (0, 1)$, any distribution $\mathcal{D}$, and any set $S = \{X^{(k)}\}_{k=1}^{m}$ of $m$ samples from $\mathcal{D}$, with probability at least $1 - \delta$, we have*

$$\left| \frac{1}{m} \sum_{k=1}^{m} H(X^{(k)}) - \mathbb{E}_{X \sim D}[H(X)] \right| = O \left( \hat{R}_m(\mathcal{H}) + \sqrt{\frac{1}{m} \log \frac{1}{\delta}} \right), \quad \text{for all} \quad H \in \mathcal{H}.$$

To bound the Rademacher complexity in our setting, we restrict to binary classification $c = 2$ and change the label space to $Y \in \{-1, 1\}^n$. For a predictor $f$, we also overload notation and let $f(z_i) \in [0, 1]$ be the output probability of node $z_i$ being classified as 1. Instead of directly using the 0-1 loss, we upper bound it using margin loss, which is defined as

$$\ell_\gamma(f(z_i), y_i) = \mathbf{1}[a_i > 0] + (1 + a_i/\gamma)\mathbf{1}\left[a_i \in [-\gamma, 0]\right]$$

where $a_i = -\tau(f(z_i), y_i) = (1 - 2f(z_i))y_i$. Then, $a_i > 0$ if and only if $z_i$ is classified incorrectly.

Now we define $H_\rho^\gamma(X) = \frac{1}{n} \sum_{i=1}^{n} \ell_\gamma(f_\rho(z_i), y_i)$ to be the margin loss of the entire graph when using a parameterized algorithm $F_\rho$. Based on this definition, we have an induced loss function family $\mathcal{H}_\rho^\gamma$. Then, given $m$ instances, for any $\gamma > 0$, we can obtain an upper bound for all $H_\rho^\gamma \in \mathcal{H}_\rho^\gamma$:

$$\mathbb{E}_{(X,Y) \sim \mathcal{D}} \left[ \frac{1}{n} \sum_{i=1}^{n} \ell_{0-1}\left(f_{\hat{\rho}}(z_i), y_i\right) \right] \leq \mathbb{E}_{(X,Y) \sim \mathcal{D}} \left[ \frac{1}{n} \sum_{i=1}^{n} \ell_\gamma\left(f_{\hat{\rho}}(z_i), y_i\right) \right] \quad \text{(by definition of } \ell_\gamma)$$

$$= \frac{1}{m} \sum_{i=1}^{m} H_\rho^\gamma(X^{(k)}) + O \left( \hat{R}_m(\mathcal{H}_\rho^\gamma) + \sqrt{\frac{\log(1/\delta)}{m}} \right).$$

$$\text{(by Theorem 2.2)}$$

Therefore, suppose we find a $\hat{\rho}$ whose empirical margin loss $1/m \sum_{i=1}^{m} H_{\hat{\rho}}^{\gamma}(X^{(k)})$ is small, and if we can show $\hat{R}_m(\mathcal{H}_{\rho}^{\gamma})$ is small, then $F_{\hat{\rho}}$ is a strong algorithm for the new problem instances. In Section 4, we bound the Rademacher complexity of graph neural network-based algorithm families.

## 3 LABEL PROPAGATION-BASED FAMILIES AND GENERALIZATION GUARANTEES

In this section, we consider three parametric families of label propagation-based algorithms, the classical type of algorithms for semi-supervised learning. Label propagation algorithms output a soft-label score $F^* \in \mathbb{R}^{n \times c}$, where the $(i, j)$-th entry of $F^*$ represents the score of class $j$ for the $i$-th sample. The prediction for the $i$-th sample is the class with the highest score, i.e. $\arg\max_{j \in [c]} F_{ij}^*$.

Below we describe each family that we considered and their corresponding pseudo-dimension bounds. Notably, the bounds for all three families of algorithms are $\Theta(\log n)$, which implies the existence of efficient algorithms with robust generalization guarantees in this setting.

### 3.1 ALGORITHM FAMILIES

We consider three parametric families, which we describe below.

**Local and Global Consistency Algorithm Family ($\mathcal{F}_\alpha$)** The first family considered is the local and global consistent algorithms Zhou et al. (2003), parameterized by $\alpha \in (0, 1)$. The optimal scoring matrix $F^*$ is defined as

$$F_\alpha^* = (1 - \alpha)(I - \alpha S)^{-1} Y, \text{ where } S = D^{-1/2} W D^{-1/2}.$$

Here, $S$ is the symmetrically normalized adjacency matrix. This score matrix $F_\alpha^*$ corresponds to minimizing the following objective function $\mathcal{Q}(F) = \frac{1}{2}(\sum_{i,j=1}^{n} W_{ij} \| \frac{1}{\sqrt{d_i}} F_i - \frac{1}{\sqrt{d_j}} F_j \|^2 + \frac{1-\alpha}{\alpha} \sum_{i=1}^{n} \| F_i - Y_i \|^2)$. The first term of $Q(F)$ measures the local consistency, i.e., the prediction between nearby points should be similar. The second term measures the global consistency, i.e., consistency to its original label. Therefore, the parameter $\alpha \in (0, 1)$ induces a trade-off between the local and the global consistency. We denote this family as $\mathcal{F}_\alpha$, and the 0-1 losses as $\mathcal{H}_\alpha$.

**Smoothing-Based Algorithm Family ($\mathcal{F}_\lambda$)** This second class of algorithm is parameterized by $\lambda \in (0, +\infty)$ (Delalleau et al., 2005). Let $\Delta \in \{0, 1\}^{n \times n}$ be a diagonal matrix where elements are 1 only if the index is in the labeled set. The scoring matrix $F_\lambda^*$ is

$$F_\lambda^* = (S + \lambda I_n \Delta_{i \in L})^{-1} \lambda Y, \text{ where} S = D - W.$$

The idea of $\mathcal{F}_\lambda$ is similar to $\mathcal{F}_\alpha$. $\lambda$ is a smoothing parameter that balances the relative importance of the known labels and the structure of the unlabeled points.

**Normalized Adjacency Matrix Based Family ($\mathcal{F}_\delta$)** Here we consider a new algorithm family which we name *Normalized Adjacency Matrix Based Family*. This class of algorithm is parameterized by $\delta \in [0, 1]$. The scoring matrix $F_\delta^*$ is

$$F_\delta^* = (I - c \cdot S)^{-1} Y, \text{ where } S = D^{-\delta} W D^{\delta - 1}.$$

Here, $S$ is the (not symmetrically) normalized adjacency matrix and $c \in \mathbb{R}$ is a constant.

This family of algorithms is motivated by $\mathcal{F}_\alpha$ and the family of spectral operators defined in Donnat & Jeong (2023). We may notice that the score matrix $F_\delta^*$ defined here is very similar to $F_\alpha^*$ in the local and global consistency family $\mathcal{F}_\alpha$ when $\alpha$ is set to a constant $c$, whose default value considered in Zhou et al. (2003) is 0.99. Here, instead of focusing on the trade-off between local and global consistency, we study the spatial convolutions $S$. With $\delta = 1$, we have the row-normalized adjacency matrix $S = D^{-1} W$. With $\delta = 0$, we have the column-normalized adjacency matrix $S = W D^{-1}$. Finally, with $\delta = 1/2$, we have the symmetrically normalized adjacency matrix that we used in $\mathcal{F}_\alpha$ and many other default implementations (Donnat & Jeong, 2023; Wu et al., 2019). We denote the set of 0-1 loss functions corresponding to $\mathcal{F}_\delta$ as $\mathcal{H}_\delta$.

## 3.2 Pseudo-dimension Guarantees

We study the generalization behavior of the three families through pseudo-dimension. The following theorems indicate that all three families have pseudo-dimension $O(\log n)$, where $n$ is the number of data in each problem instance. This result suggests that, all three families of algorithms require $m = O\left(\log n/\epsilon\right)$ problem instances to learn a $\epsilon$-optimal algorithmic parameter. We also complement our upper bounds with matching pseudo-dimension lower bound $\Omega(\log n)$, which indicates that we cannot always learn a near-optimal parameter if the number of problem instances is further reduced.

**Theorem 3.1.** *The pseudo-dimension of the Local and Global Consistency Algorithmic Family, $\mathcal{F}_\alpha$, is $\mathrm{PDIM}(\mathcal{H}_\alpha) = \Theta(\log n)$, where $n$ is the total number of labeled and unlabeled data points.*

**Theorem 3.2.** *The pseudo-dimension of the Smoothing-Based Algorithmic Family, $\mathcal{F}_\lambda$, is $\mathrm{PDIM}(\mathcal{H}_\lambda) = \Theta(\log n)$, where $n$ is the total number of labeled and unlabeled data points.*

**Theorem 3.3.** *The pseudo-dimension of the Normalized Adjacency Matrix-Based Algorithmic Family, $\mathcal{F}_\delta$, is $\mathrm{PDIM}(\mathcal{H}_\delta) = \Theta(\log n)$, where $n$ is the total number of labeled and unlabeled data points.*

The proofs of the above three theorems follow a similar template. Here, we give an overview of the proof idea. The full proof is in Appendix A.

**Upper Bound** First, we investigate the function structure of each index in $F^*$. For the function classes $\mathcal{F}_\alpha$ and $\mathcal{F}_\lambda$, the following lemma is useful.

**Lemma 3.4.** *Let $A, B \in \mathbb{R}^{n \times n}$, and $C(x) = (A + xB)^{-1}$ for some $x \in \mathbb{R}$. Each entry of $C(x)$ is a rational polynomial $P_{ij}(x)/Q(x)$ for $i, j \in [n]$ with each $P_{ij}$ of degree at most $n-1$ and $Q$ of degree at most $n$.*

This lemma reduces each index in the matrix of form $C(x) = (A + xB)^{-1}$ into a polynomial of parameter $x$ with degree at most $n$. By definition, we can apply this lemma to $F_\alpha^*$ and $F_\lambda^*$ and conclude that each index of these matrices is a degree-$n$ polynomial of variable $\alpha$ and $\lambda$, respectively.

For the algorithm family $\mathcal{F}_\delta$, the following lemma is helpful:

**Lemma 3.5.** *Let $S = D^{-x}WD^{x-1} \in \mathbb{R}^{n \times n}$, and $C(x) = (I - c \cdot S)^{-1}$ for some constant $c \in (0, 1)$ and variable $x \in [0, 1]$. For any $i, j \in [n]$, the $i, j$-the entry of $C(x)$ is an exponential $C(x)_{ij} = a_{ij} \exp(b_{ij}x)$ for some constants $a_{ij}, b_{ij}$.*

By definition of $F_\delta^*$, this lemma indicates that each index of $F_\delta^*$ is a weighted sum of $n$ exponentials of the hyperparameter $\delta$.

For $F^*$ being a prediction matrix of any of the above three algorithmic family, recall that the prediction on each node $i \in [n]$ is $\hat{y}_i = \mathrm{argmax}_{j \in [c]}([F^*]_{ij})$, so the prediction on a node can change only when $\mathrm{sign}([F^*]_{ij} - [F^*]_{ik})$ changes for some classes $j, k \in [c]$. For the families $\mathcal{F}_\alpha$ and $\mathcal{F}_\lambda$, $[F^*]_{ij} - [F^*]_{ik}$ is a rational polynomial $(P_{ij}(\alpha) - P_{ik}(\alpha))/Q(\alpha)$, where $(P_{ij}(\alpha) - P_{ik}(\alpha))$ and $Q(\alpha)$ are degree of at most $n$ (we can simply replace $\alpha$ with $\lambda$ for $\mathcal{F}_\lambda$). Therefore, its sign can only change at most $O(n)$ times. For the family $\mathcal{F}_\delta$, we refer to the following lemma and conclude that the sign of $F_{ij}^* - F_{ik}^*$ can only change at most $O(n)$ times as well.

**Lemma 3.6.** *Let $a_1, \ldots, a_n \in \mathbb{R}$ be not all zero, $b_1, \ldots, b_n \in \mathbb{R}$, and $f(x) = \sum_{i=1}^{n} a_i e^{b_i x}$. The number of roots of $f$ is at most $n - 1$.*

Therefore, for all three families, the prediction on a single node can change at most $\binom{c}{2}O(n) \in O(nc^2)$ times as the hyperparameter varies. For $m$ problem instances, each of $n$ nodes, this implies we have at most $O(mn^2c^2)$ distinct values of the loss function. The pseudo-dimension $m$ then satisfies $2^m \le O(mn^2c^2)$, which implies $\mathrm{PDIM}(\mathcal{H}_\alpha) = \mathrm{PDIM}(\mathcal{H}_\lambda) = \mathrm{PDIM}(\mathcal{H}_\delta) = O(\log n)$.

**Lower Bound** Our proof relies on a collection of parameter thresholds and well-designed labeling instances that are shattered by the thresholds. Here we present the proof idea of pseudo-dimension lower bound of the family $\mathcal{F}_\alpha$. The analysis for $\mathcal{F}_\lambda$ and $\mathcal{F}_\delta$ depends on a similar construction.

We first describe a hard instance of $4$ nodes, using binary labels $a$ and $b$. We have two points labeled $a$ (namely $a_1, a_2$), and one point labeled $b$ (namely $b_1$) connected with both $a_1$ and $a_2$ with edge

weight 1. We also have an unlabeled point $u$ connected to $b_1$ with edge weight $x \geq 0$. That is, the affinity matrix and initial labels are

$$W = \begin{bmatrix} 0 & 1 & 1 & x \\ 1 & 0 & 0 & 0 \\ 1 & 0 & 0 & 0 \\ x & 0 & 0 & 0 \end{bmatrix}, Y = \begin{bmatrix} 1 & 0 \\ 0 & 1 \\ 0 & 1 \\ 0 & 0 \end{bmatrix}.$$

With this construction, the prediction on node $u$ changes and only change when $\alpha = \frac{(x+2)^{1/2}}{2}$. For any $\beta \in [0, 1]$ and let $x = 4\beta^2 - 2 \geq 0$, then $\hat{y}_4 = 0$ when $\alpha < \beta$ and $\hat{y}_4 = 1$ when $\alpha \geq \beta$.

Now we can create a large graph of $n$ nodes, consisting of $n/4$ connected components as described above. We assume 4 divides $n$ for simplicity. Given a sequence of $\alpha$'s such that $0 < \alpha_0 < 1/\sqrt{2} \leq \alpha_1 < \alpha_2 < ... < \alpha_{n/4} < 1$, we can create the $i$-th connected component with $x = 4\alpha_i^2 - 2$. Now the predicted label of the unlabeled node in the $i$-th connected component is 0 when $\alpha < \alpha_i$ and 1 when $\alpha \geq \alpha_i$. By alternatively labeling these unlabeled nodes, the 0-1 loss of this problem instance fluctuates as $\alpha$ increases.

Finally, by precisely choosing the subsequences so that the oscillations align with the bit flips in the binary digit sequence, we can construct m instances that satisfy the $2^m$ shattering constraints.

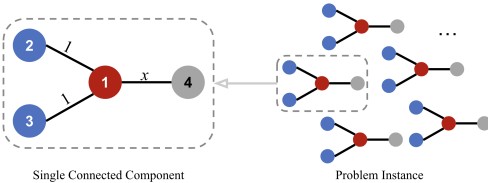

Single Connected Component      Problem Instance

Figure 1: An illustration of the construction of the problem instance in the lower bound proof.

**Remark 1.** *We reiterate the implications of the above three theorems. All three families have pseudo-dimension $\Theta(\log n)$. This indicates that all three families of algorithms require $m = O\left(\log n / \epsilon\right)$ problem instances to learn a $\epsilon$-optimal hyperparameter.*

## 4    GNN FAMILIES AND THEIR GENERALIZATION GUARANTEES

In this section, we study hyperparameter selection for Graph Neural Networks (GNNs) (Kipf & Welling, 2017; Veličković, Petar et al., 2018; Iscen et al., 2019), which excel in tasks involving graph-structured data like social networks, recommendation systems, and citation networks. To understand generalization in hyperparameter selection for GNNs, we analyze Rademacher complexity.

To the best of our knowledge, we are the first to provide generalization guarantees for hyperparameter selection. Prior work (Garg et al., 2020) focused on Rademacher complexity for graph classification with fixed hyperparameters, whereas we address node classification across multiple instances, optimizing hyperparameters.

In Section 4.1, we examine the Rademacher complexity bound of a basic Simplified Graph Convolutional Network (Wu et al., 2019) family, as a foundation for the more complex family.

In Section 4.2, we introduce a novel architecture, which we call GCAN, that uses a hyperparameter $\eta \in [0, 1]$ to interpolate two popular GNNS: the graph convolutional neural networks (GCN) and graph attention neural networks (GAT). GCAN selects the optimal model for specific datasets: $\eta = 0$ corresponds to GCN, $\eta = 1$ to GAT, and intermediate values explore hybrid architectures that may outperform both. We also establish a Rademacher complexity bound for the GCAN family.

Our proofs for SGC and GCAN share a common strategy: modeling the 0-1 loss of each problem instance as an aggregation of single-node losses, reducing the problem to bounding the Rademacher complexity of computation trees for individual nodes. Specifically, we upper bound the 0-1 loss with a margin loss, then relate the complexity of problem instances to the computation trees of nodes. Using a covering argument, we bound the complexity of these trees by analyzing margin loss changes due to parameter variations.

For each node $z_i$, we define its computation tree of depth $L$ to represent the structured $L$-hop neighborhood of $v$, where the children of any node $u$ are the neighbors of $u$, $\mathcal{N}_u$. Denote the computation tree of $z_i$ as $t_i$, and the learned parameter as $\theta$, then $l_\gamma(z_i) = l_\gamma(t_i, \theta)$. We can now rewrite $l_\gamma(Z)$ as an expectation over functions applied to computation trees. Let $t_1, ..., t_t$ be the set of all possible computation trees of depth $L$, and $w_i(Z)$ the number of times $t_i$ occurs in $Z$. Then, we have

$$l_\gamma(Z) = \sum_{i=1}^{t} \frac{w_i(Z)}{\sum_{j=1}^{t} w_j(Z)} l_\gamma(t_i, \theta) = \mathbb{E}_{t \sim w'(Z)} l_\gamma(t, \theta).$$

The following proposition indicates that it suffices to bound the Rademacher Complexity of single-node computation trees.

**Proposition 4.1** (Proposition 6 from Garg et al. (2020).)**.** *Let $S = \{Z_1, ..., Z_m\}$ be a set of i.i.d. graphs, and let $\mathcal{T} = \{t_1, ..., t_m\}$ be such that $t_j \sim w'(Z_j), j \in [m]$. Denote by $\hat{R}_S$ and $\hat{R}_\mathcal{T}$ the empirical Rademacher complexity of $\mathcal{H}_\rho^\gamma s$ for graphs $S$ and trees $\mathcal{T}$. Then, $\hat{R}_S = \mathbb{E}_{t_1, ..., t_m} \hat{R}_\mathcal{T}$.*

### 4.1 SIMPLIFIED GRAPH CONVOLUTIONAL NETWORK FAMILY

Simplified Graph Convolution Network (SGC) is introduced by Wu et al. (2019). By removing non-linearities and collapsing weight matrices between consecutive layers, SGC reduces the complexity of GCN while maintaining high accuracy in many applications.

Consider input data $X = (n, Z, L, G)$, where the feature is written as a matrix $Z \in \mathbb{R}^{n \times d}$. For any value of the hyperparameter $\beta \in [0, 1]$, let $\tilde{W} = W + \beta I$ be the augmented adjacency matrix, $\tilde{D} = D + \beta I$ be the corresponding degree matrix, and $S = \tilde{D}^{-1/2} \tilde{W} \tilde{D}^{-1/2}$ be the normalized adjacency matrix. Let $\theta \in \mathbb{R}^d$ be the learned parameter. The SGC classifier of depth $L$ is

$$\hat{Y} = \text{softmax}(S^L Z \theta).$$

We focus on learning the algorithm hyperparameter $\beta \in [0, 1]$ and define the SGC algorithm family as $\mathcal{F}_\beta$. We denote the class of margin losses induced by $\mathcal{F}_\beta$ as $\mathcal{H}_\beta^\gamma$. To study the generalization ability to tune $\beta$, we bound the Rademacher complexity of $\mathcal{H}_\beta^\gamma$. The proof is detailed in Appendix C.1.

**Theorem 4.2.** *Assuming $D, W$, and $Z$ are bounded (the assumptions in Bartlett et al. (2017); Garg et al. (2020)), i.e. $d_i \in [C_{dl}, C_{dh}] \subset \mathbb{R}^+$, $w_{ij} \in [0, C_w]$, and $\|Z\| \leq C_z$, we have that the Rademacher complexity of $\mathcal{H}_\beta^\gamma$ is bounded:*

$$\hat{R}_m(\mathcal{H}_\beta^\gamma) = O\left(\frac{\sqrt{dL \log \frac{C_{dh}}{C_{dl}} + d \log \frac{mC_z C_\theta}{\gamma}}}{\sqrt{m}}\right).$$

This theorem indicates that the number of problem instances needed to learn a near-optimal hyperparameter only scale polynomially with the input feature dimension $d$ and the number of layers $L$ of the neural networks, and only scales logarithmically with the norm bounds $C$'s and the margin $\gamma$.

### 4.2 GCAN INTERPOLATION AND ITS RADEMACHER COMPLEXITY

In practice, GCN and GAT outperform each other in different problem instances (Dwivedi et al., 2023). To effectively choose the better algorithm, we introduce a family of algorithms that *interpolates* GCN and GAT, parameterized by $\eta \in [0, 1]$. This family includes both GCN and GAT, so by choosing the best algorithm within this family, we can automatically select the better algorithm of the two, specifically for each input data. Moreover, GCAN could potentially outperform both GAT and GCN by taking $\eta$ as values other than $0$ and $1$. We believe such an interpolation technique could potentially be used to select between other algorithms that share similar architecture.

Recall that in both GAT and GCN, the update equation has the form of activation and a summation over the feature of all neighboring vertices in the graph (a brief description of GAT and GCN is given in Appendix B). Thus, we can interpolate between the two update rules by introducing a

hyperparameter $\eta \in [0, 1]$, where $\eta = 0$ corresponds to GCN and $\eta = 1$ corresponds to GAT. Formally, given input $X = (n, \{z_i\}_{i=1}^n, L, G)$, we initialize $h_i^0 = z_i$ and update at a level $\ell$ by

$$h_i^\ell = \sigma \left( \sum_{j \in \mathcal{N}_i} \left( \eta \cdot e_{ij}^\ell + (1 - \eta) \cdot \frac{1}{\sqrt{d_i d_j}} \right) U^\ell h_j^\ell \right)$$

$$\text{where } e_{ij}^\ell = \frac{\exp(\hat{e}_{ij}^\ell)}{\sum_{j' \in \mathcal{N}_i} \exp(\hat{e}_{ij'}^\ell)}, \quad \hat{e}_{ij}^\ell = \sigma(V^\ell [U^\ell h_i^\ell, U^\ell h_j^\ell]).$$

Here $e_{ij}^\ell$ is the attention score of node $j$ for node $i$. $V^\ell$ and $U^\ell$ are learnable parameters. $\sigma(\cdot)$ is a 1-Lipschitz activation function (e.g. ReLU, sigmoid, etc.). $[U^\ell h_i^\ell, U^\ell h_j^\ell]$ is the concatenation of $U^\ell h_i^\ell$ and $U^\ell h_j^\ell$. We denote this algorithm family by $\mathcal{F}_\eta$ and the induced margin loss class by $\mathcal{H}_\eta^\gamma$.

While our primary focus is not the comparative performance of GCAN against GAT or GCN, our curiosity led us to conduct additional experiments, presented in Appendix D. The results consistently show that GCAN matches or exceeds the performance of both GAT and GCN.

**Theorem 4.3.** *Assume the parameter $U^\ell$ is shared over all layers, i.e. $U^\ell = U$ for all $\ell \in [L]$ (the assumption used in Garg et al. (2020)), and the parameters are bounded: $\|U\|_F \leq C_U$, $\|V^\ell\|_2 \leq C_V$, $\|z_i\| \leq C_z$, and $d_i \in [C_{dl}, C_{dh}]$. Denoting the branching factor by $r = \max_{i \in [n]} |\sum_{j \in [n]} \mathbb{I}[w_{ij} \neq 0]|$, we have that the Rademacher complexity of $\mathcal{H}_\eta^\gamma$ is bounded:*

$$\hat{R}_m(\mathcal{H}_\eta^\gamma) = O \left( \frac{d\sqrt{L \log \frac{rC_U}{C_{dl}+C_U} + \log \frac{mdC_z}{\gamma}}}{\sqrt{m}} \right).$$

The proof of Theorem 4.3 is similar to that of Theorem 4.2. See Appendix C.2 for details.

**Remark 2.** *The main difference between the Rademacher Complexity of Simplified Graph Convolution Network (Theorem 4.2) and GCAN (Theorem 4.3) is the dependency on feature dimension $d$: $\sqrt{d}$ for SGC and $d$ for GCAN. This difference arises from the dimensionality of the parameters. The parameter $\theta$ in SGC has dimension $d$, but the parameter $U$ and $V$ in GCAN have dimension $d \times d$ and $1 \times 2d$, respectively. As GCAN is a richer model, it requires more samples to learn, but this is not a drawback; its complexity allows it to outperform SGC in many scenarios.*

**Remark 3.** *There are no direct dependencies on $n$ in Theorem 4.2 and Theorem 4.3, but the dependency is implicitly captured by the more fine-grained value $C_{dl}, C_{dh}$, and $C_Z$. Here, $C_{dl}$ and $C_{dh}$ are the lower and upper bounds of the degree (number of neighbors) of the nodes, which generally increase with $n$. $C_Z$ is the Frobenius norm of the feature matrix $Z \in \mathbb{R}^{n \times d}$. Since the size of $Z$ scales with $n$, the value of $C_Z$ is generally larger for larger $n$.*

## 5 EXPERIMENTS

In this section, we empirically verify the effectiveness of our hyperparameter selection method.

### 5.1 LABEL PROPAGATION-BASED METHOD: NORMALIZED ADJACENCY MATRIX-BASED ALGORITHMIC FAMILY

We empirically validate our findings in Section 3. For each of the eight datasets, the number of nodes per problem instance, $n$, is fixed at 30. We set the target generalization error to $\epsilon = 0.01$, and calculate the required number of problem instances as $m = O(\log n/\epsilon) \approx 300$. To evaluate performance, we randomly sample 300 graphs with 30 nodes each, tune the hyperparameter values to maximize accuracy on these graphs, and then test the selected hyperparameter on a separate set of 300 randomly sampled graphs. The results of evaluating the Normalized Adjacency Matrix-Based Algorithmic Family is presented in Table 1, confirming that the observed generalization error is within the scale of the target value 0.01.

### 5.2 GNN-BASED METHOD: GCAN EXPERIMENT

The bounds provided in Theorem 4.2 and Theorem 4.3 involve multiple constants, making them challenging to compute directly. Therefore, we focus on demonstrating the effectiveness of our

|            | CIFAR10 | WikiCS | CORA   | Citeseer | PubMed | AmazonPhotos | Actor  |
|------------|---------|--------|--------|----------|--------|--------------|--------|
| Train Acc. | 0.9445  | 0.7522 | 0.7927 | 0.7845   | 0.9993 | 0.9983       | 0.9185 |
| Test Acc.  | 0.9397  | 0.7485 | 0.8010 | 0.7714   | 0.9993 | 0.9989       | 0.9239 |
| Abs. Diff. | 0.0048  | 0.0037 | 0.0083 | 0.0131   | 0.     | 0.0006       | 0.0054 |

Table 1: The Training Accuracy and Testing Accuracy of learning the hyperparameter $\delta$ in Normalized Adjacency Matrix Based Family ($\mathcal{F}_\delta$). The absolute difference between the accuracies (i.e. generalization error) is within the scale of our target value $0.01$.

approach for selecting algorithm hyperparameters in our setup. To illustrate this, we compare the performance of GCAN with tuned hyperparameters against GAT and GCN.

For each dataset, we sample 20 random sub-graphs of 100 nodes to learn the optimal hyperparameter $\eta$ via backpropagation. A large disconnected graph is formed by combining these sub-graphs, allowing parameter values to vary across graphs while sharing a unified learnable $\eta$. The optimized hyperparameter is then tested on another 20 sub-graphs from the dataset

The results are shown in Figure 2. It is evident that GCAN consistently achieves higher or comparable accuracy compared to both GAT and GCN across all datasets. Notably, GCAN demonstrates significant improvements in CIFAR10 and CORA, highlighting its effectiveness in these scenarios.

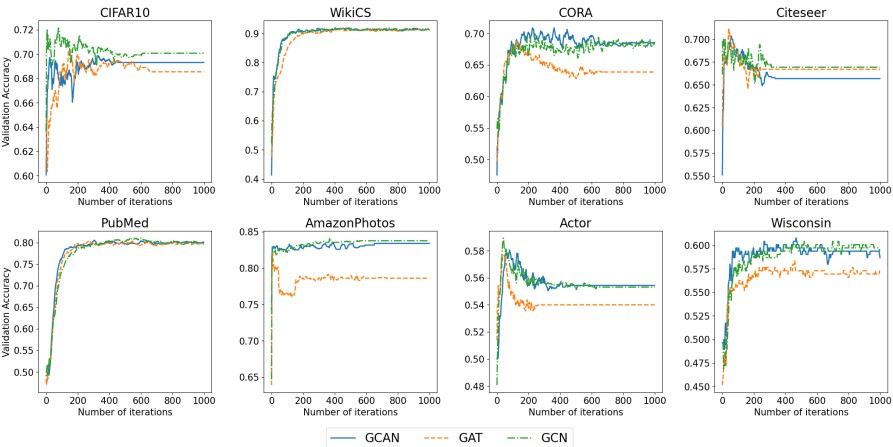

Figure 2: Validation Accuracy (computed on the unlabeled nodes across 20 testing graphs) vs. iterations. GCAN shows no worse accuracy when compared with both GAT and GCN.

## 6 CONCLUSION

We study algorithm selection in graph-based semi-supervised learning, by tuning real-valued hyperparameters that define algorithmic families. Our approach can improve the accuracy of prediction in semi-supervised learning by selecting the most effective data-specific algorithmic hyperparameter automatically. We do this by leveraging access to multiple instances of data from a given domain and providing formal guarantees on the number of data samples needed to learn the best algorithm for several classical parameterized families as well as a novel family that interpolates convolution (GCN) and attention (GAT) in graph neural networks.

Our work also opens up several interesting directions for future research. For the graph neural network hyperparameter tuning, we only upper bound the Rademacher Complexity, and it would be interesting to obtain lower bounds and determine the tightness of our bounds. We also expect our techniques to be applicable to other GNN architectures and graph-based semi-supervised learning families in the literature. Moreover, we only consider single real-valued hyperparameter in our work, and it would be interesting to investigate the generalizability of learning multiple hyperparamerters with our approach. Lastly, we limit ourselves on the sample complexity, it is an interesting question to develop effective and computationally efficient implementations for learning the hyperparameters.

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

APPENDIX

# A    PROOFS IN SECTION 3

We provide additional proof details from Section 3 below.

## A.1    PROOF OF LEMMA 3.4

*Proof.* Using the adjugate matrix, we have

$$C(x) = \frac{1}{\det(A + xB)} \text{adj}(A + xB).$$

The determinant of $A + xB$ can be written as

$$\det(A + xB) = \sum_{\sigma \in S_n} \left( \text{sgn}(\sigma) \prod_{i=1}^{n} [A + xB]_{i\sigma_i} \right),$$

where $S_n$ represents the symmetric group and $\text{sgn}(\sigma) \in \{\pm 1\}$ is the signature of permutation $\sigma$. Thus $\det(A + xB)$ is a polynomial of $x$ with a degree at most $n$. The adjugate of $A + xB$ is

$$\text{adj}(A + xB) = C^\top,$$

where $C$ is the cofactor matrix of $A + xB$. By definition, each entry of $C$ is $C_{ij} = (-1)^{i+j} k_{ij}$ where $k_{ij}$ is the determinant of the $(n-1) \times (n-1)$ matrix that results from deleting $i$-th row and $j$-th column of $A + xB$. This implies that each entry of $C$ (and thus $\text{adj}(A + xB)$) is a polynomial of degree at most $n - 1$. Letting $Q(x) = \det(A + xB)$ and $P_{ij}(x) = [\text{adj}(A + xB)]_{ij}$ concludes our proof. □

## A.2    PROOF OF LEMMA 3.5

*Proof.* The $ij$-th element of $I - c \cdot S$ is

$$[I - c \cdot S]_{ij} = \begin{cases} -c \cdot d_i^{-\delta} W_{ij} d_j^{\delta-1} = -(d_i^{-1} d_j)^\delta (c \cdot W_{ij} d_j^{-1}) & \text{, if } i \neq j \\ 1 = (d_i^{-1} d_i)^\delta & \text{, otherwise.} \end{cases}$$

Using adjugate matrix, we have

$$(I - c \cdot S)^{-1} = \frac{1}{\det(I - c \cdot S)} \text{adj}(I - c \cdot S).$$

Note that the determinant of any $k \times k$ matrix $A$ can be written as

$$\det(A) = \sum_{\sigma \in S_k} \left( \text{sgn}(\sigma) \prod_{i=1}^{k} [A]_{i\sigma_i} \right),$$

where $S_k$ represents the symmetric group and $\text{sgn}(\sigma) \in \{\pm 1\}$ is the signature of permutation $\sigma$.

Now consider $\text{adj}(I - c \cdot S)$. Let $M_{ij}$ be the $(n-1) \times (n-1)$ matrix resulting from deleting $i$-th row and $j$-th column from $[I - c \cdot S]$. Then,

$$[\text{adj}(I - c \cdot S)]_{ij} = (-1)^{i+j} \det(M_{ji}) = \sum_{\sigma \in S_{n-1}} \left( \text{sgn}(\sigma) \prod_{k=1}^{n-1} [M_{ji}]_{k\sigma_k} \right) = \sum_{\sigma \in S_{n-1}} (a_\sigma \exp(\delta \ln b_\sigma)),$$

for some constants $a_\sigma, b_\sigma$ that satisfies

$$b_\sigma = ( \prod_{k \in [n] \setminus \{j\}} d_k^{-1})( \prod_{k \in [n] \setminus \{i\}} d_k) = d_i^{-1} d_j.$$

We can then rewrite $[\text{adj}(I - c \cdot S)]_{ij}$ as

$$[\text{adj}(I - c \cdot S)]_{ij} = \sum_{\sigma \in S_{n-1}} (a_\sigma \exp(\delta \ln(d_i^{-1} d_j))) = a_{ij} \exp(\delta \ln(d_i^{-1} d_j)),$$

where $a_{ij} = \sum_{\sigma \in S_{n-1}} a_\sigma$. □

### A.3 PROOF OF LEMMA 3.6

*Proof.* We prove by induction on $n$. If $n = 1$, then $f(x) = ae^{bx}$ and $a \neq 0$, so $f(x)$ has $0 = n - 1$ root. Now assume that the statement holds for some $n = m$ and consider when $n = m + 1$. That is, we have

$$f(x) = \sum_{i=1}^{m+1} a_i e^{b_i x}.$$

Assume for the sake of contradiction that $f$ has $n = m + 1$ roots. Define

$$g(x) = \frac{f(x)}{e^{b_{m+1}x}} = \sum_{i=1}^{m} a_i e^{(b_i - b_{m+1})x} + a_{m+1},$$

then $g$ also has $m + 1$ roots. Since $g$ is continuous,

$$g'(x) = \sum_{i=1}^{m} (b_i - b_{m+1}) a_i e^{(b_i - b_{m+1})x}$$

must have $m$ roots. However, using our induction hypothesis, it should have at most $m - 1$ roots. This means our assumption is incorrect, i.e. $f$ must have at most $m = n - 1$ roots.

We conclude that $f$ must have at most $n - 1$ roots. $\qquad\square$

### A.4 PROOF OF THEOREM 3.1

**Upper Bound.** Proof is given in Section 3.

**Lower Bound.** We first construct the small connected component of 4 nodes:

**Lemma A.1.** *Given $x \in [1/\sqrt{2}, 1)$, there exists a labeling instance $(G, L)$ with 4 nodes, such that the predicted label of the unlabeled points changes only at $\alpha = x$ as $\alpha$ varies in $(0, 1)$.*

*Proof.* We use binary labeling $a$ and $b$. We have two points labeled $a$ (namely $a_1, a_2$), and one point labeled $b$ (namely $b_1$) connected with both $a_1$ and $a_2$ with edge weight 1. We also have an unlabeled point $u$ connected to $b_1$ with edge weight $x \geq 0$. That is, the affinity matrix and initial labels are

$$W = \begin{bmatrix} 0 & 1 & 1 & x \\ 1 & 0 & 0 & 0 \\ 1 & 0 & 0 & 0 \\ x & 0 & 0 & 0 \end{bmatrix}, Y = \begin{bmatrix} 1 & 0 \\ 0 & 1 \\ 0 & 1 \\ 0 & 0 \end{bmatrix}.$$

Recall that the score matrix is

$$F^* = (1 - \alpha)(I - \alpha S)^{-1} Y.$$

We now calculate:

$$D^{-1/2} = \begin{bmatrix} (x+2)^{-1/2} & 0 & 0 & 0 \\ 0 & 1 & 0 & 0 \\ 0 & 0 & 1 & 0 \\ 0 & 0 & 0 & x^{-1/2} \end{bmatrix},$$

$$S = D^{-1/2} W D^{-1/2} = \begin{bmatrix} 0 & (x+2)^{-1/2} & (x+2)^{-1/2} & x^{1/2}(x+2)^{-1/2} \\ (x+2)^{-1/2} & 0 & 0 & 0 \\ (x+2)^{-1/2} & 0 & 0 & 0 \\ x^{1/2}(x+2)^{-1/2} & 0 & 0 & 0 \end{bmatrix},$$

$$(I - \alpha S)^{-1} = \frac{1}{\det(I - \alpha S)} \text{adj}(I - \alpha S)$$

$$= \frac{1}{1 - \alpha^2} \begin{bmatrix} 1 & \frac{\alpha}{(x+2)^{1/2}} & \frac{\alpha}{(x+2)^{1/2}} & \frac{\alpha x^{1/2}}{(x+2)^{1/2}} \\ \frac{\alpha}{(x+2)^{1/2}} & 1 - \frac{\alpha^2(x+1)x}{(x+2)} & \frac{\alpha^2}{x+2} & \frac{\alpha^2 x^{1/2}}{(x+2)} \\ \frac{\alpha}{(x+2)^{1/2}} & \frac{\alpha^2}{x+2} & 1 - \frac{\alpha^2(x+1)x}{(x+2)} & \frac{\alpha^2 x^{1/2}}{(x+2)} \\ \frac{\alpha x^{1/2}}{(x+2)^{1/2}} & \frac{\alpha^2 x^{1/2}}{(x+2)} & \frac{\alpha^2 x^{1/2}}{(x+2)} & 1 - \frac{2\alpha^2}{x+2} \end{bmatrix}.$$

Recall that the prediction on the unlabeled point is $\hat{y}_4 = \text{argmax} F_4^*$, so we calculate

$$
\begin{aligned}
\hat{y}_4 = \text{sign}(F_{4,2}^* - F_{4,1}^*) &= \text{sign}\left(\frac{\alpha x^{1/2}(2\alpha - (x+2)^{1/2})}{(1+\alpha)(x+2)}\right) \\
&= \text{sign}\left(x^{1/2}(2\alpha - (x+2)^{1/2})\right). \qquad \text{(since } \alpha \in (0,1) \text{ and } x \geq 0)
\end{aligned}
$$

Solving the equation $x^{1/2}(2\alpha - (x+2)^{1/2}) = 0$, we know that the prediction changes and only change when $\alpha = \frac{(x+2)^{1/2}}{2}$. Let $x = 4x^2 - 2 \geq 0$, then $\hat{y}_4 = 0$ when $\alpha < x$ and $\hat{y}_4 = 1$ when $\alpha \geq x$, which completes our proof. $\qquad \square$

**Lemma A.2.** *Given integer $n > 1$ and a sequence of $\alpha$'s such that $0 < \alpha_0 < 1/\sqrt{2} \leq \alpha_1 < \alpha_2 < ... < \alpha_n < 1$, there exists a real-valued witness $w > 0$ and a problem instance of partially labeled $4n$ points, such that for $0 \leq i \leq n/2 - 1$, $l < w$ for $\alpha \in (\alpha_{2i}, \alpha_{2i+1})$, and $l > w$ for $\alpha \in (\alpha_{2i+1}, \alpha_{2i+2})$.*

*Proof.* We create $n$ connected components using the previous lemma, with $x_i = \alpha_i$. Let the unlabeled point in the $i$th component be $u_i$, then as $\alpha$ increases from $\alpha_{i-1}$ to $\alpha_i$, the predicted label of $u_i$ changes from $a$ to $b$. If the sequence $u_i$ is alternately labeled with $u_1$ labeled $a$, then the loss increases and decreases alternately as all the labels turn to $b$ when $\alpha$ increases to $\alpha_n$. Specifically, as $\alpha$ increases to $\alpha_1$, the point $u_1$ has predicted label changes from $a$ to $b$. Since its true label is $a$ and the predicted labels of other $u_i$'s remain unchanged, our loss slightly increases to $l_{max}$. Then, as $\alpha$ increases to $\alpha_2$, the point $u_2$ gets correctly labeled as $b$ and all other nodes unchanged, which slightly decreases our loss back to $l_{min}$. The loss thus fluctuates between $l_{min}$ and $l_{max}$. We therefore set the witness $w$ as something in between.

$$
w = \frac{l_{min} + l_{max}}{2}.
$$

$\qquad \square$

We now finish the lower bound proof for Theorem 3.1.

*Proof.* Arbitrarily choose $n' = n/4$ (assumed to be a power of 2 for convenient representation) real numbers $1/\sqrt{2} \leq \alpha_{[000..1]} < \alpha_{[000...10]} < ... < \alpha_{[111...11]} < 1$. The indices are increasing binary numbers of length $m = \log n'$. We create $m$ labeling instances that can be shattered by these $\alpha$ values. For the $i$-th instance $(X^{(i)}, Y^{(i)})$, we apply the previous lemma with a subset of the $\alpha_b$ sequence that corresponds to the $i$-th bit flip in $b$, where $b \in \{0,1\}^m$. For example, $(X^{(1)}, Y^{(1)})$ is constructed using $r_{[100..0]}$, and $(X^{(2)}, Y^{(2)})$ is constructed using $r_{[010..0]}, r_{[100.0]}$ and $r_{[110..0]}$. The lemma gives us both the instances and the sequence of witnesses $w_i$.

This construction ensures $\text{sign}(l_{\alpha_b} - w_i) = b_i$ for all $b \in \{0,1\}^m$. Thus the pseudo-dimension is at least $\log n' = \log n - \log 4 = \Omega(\log n)$ $\qquad \square$

### A.5 PROOF OF THEOREM 3.2

**Upper Bound.** The closed-form solution $F^*$ is given by

$$
F^* = (S + \lambda I_n \Delta_{i \in L})^{-1} \lambda Y.
$$

By Lemma 3.4, each coefficient $[F^*]_{ij}$ is a rational polynomial in $\lambda$ of the form $P_{ij}(\lambda)/Q(\lambda)$ where $P_{ij}$ and $Q$ are polynomials of degree $n$ and $n$ respectively. Note that the prediction for each node $i \in [n]$ is $\hat{y}_i = \arg\max_{j \in c} f_{ij}$ and thus the prediction on any node in the graph can only change when $sign(f_{ij} - f_{ik})$ changes for some $j, k \in [c]$. Note that $f_{ij} - f_{ik}$ is also a rational polynomial $(P_{ij}(\lambda) - P_{ik}(\lambda))/Q(\lambda)$ where both the numerator and denominator are polynomials in $\lambda$ of degree $n$, meaning the sign can change at most $O(n)$ times. As we vary $\lambda$, we have that the prediction on a single node can change at most $\binom{c}{2}O(n) \in O(nc^2)$. Across the $m$ problem instances and the $n$ total nodes, we have at most $O(n^2 c^2 m)$ distinct values of our loss function. The pseudo-dimension $m$ thus satisfies $2^m \leq O(n^2 c^2 m)$, or $m = O(\log n)$

**Lower Bound.** We construct the small connected component of $4$ nodes as follows:

**Lemma A.3.** *Given $\lambda' \in (1, \infty)$, there exists a labeling instance $(X, Y)$ with $4$ nodes, such that the predicted label of the unlabeled points changes only at $\lambda = \lambda'$ as $\lambda$ varies in $(0, \infty)$.*

*Proof.* We use binary labeling $a$ and $b$. We have two points labeled $a$ (namely $a_1, a_2$), and one point labeled $b$ (namely $b_1$). We also have an unlabeled point $u$ connected to $b_1$ with edge weight $x \geq 0$ and connected with both $a_1$ and $a_2$ with edge weight $1$. That is, the weight matrix and initial labels are

$$
W = \begin{bmatrix} 0 & 0 & 1 & 0 \\ 0 & 0 & 1 & 0 \\ 1 & 1 & 0 & x \\ 0 & 0 & x & 0 \end{bmatrix}, Y = \begin{bmatrix} -1 \\ -1 \\ 0 \\ 1 \end{bmatrix}.
$$

The closed form solution is

$$
F^* = (S + \lambda I_n \Delta_{i \in L})^{-1} \lambda Y
$$

where $S = \mathrm{diag}(W \vec{1}_n) - W$. We now calculate:

$$
S = \begin{bmatrix} 1 & 0 & -1 & 0 \\ 0 & 1 & -1 & 0 \\ -1 & -1 & x+2 & -x \\ 0 & 0 & -x & x \end{bmatrix}
$$

$$
S + \lambda I_n \Delta_{i \in L} = \begin{bmatrix} 1+\lambda & 0 & -1 & 0 \\ 0 & 1+\lambda & -1 & 0 \\ -1 & -1 & x+2 & -x \\ 0 & 0 & -x & x+\lambda \end{bmatrix}
$$

Recall that the prediction on the unlabeled point is $\hat{y}_3 = \mathrm{sign}([F*]_{32} - [F^*]_{31})$, so we calculate

$$
\hat{y}_3 = \mathrm{sign}(F*]_{32} - [F^*]_{31}) = \mathrm{sign}\left( -2\lambda \left( \frac{\lambda + x}{\lambda^2 x + 2\lambda^2 + 3\lambda x} \right) + \lambda \left( \frac{\lambda x + x}{\lambda^2 x + 2\lambda^2 + 3\lambda x} \right) \right)
$$

$$
= \mathrm{sign}\left( -2\lambda(\lambda + x) + \lambda(\lambda x + x) \right) \qquad \text{(since } \lambda > 0 \text{ and } x \geq 0)
$$

$$
= \mathrm{sign}\left( -2(\lambda + x) + (\lambda x + x) \right) \qquad \text{(since } \lambda > 0)
$$

$$
= \mathrm{sign}\left( -2\lambda - x + \lambda x \right)
$$

Solving the equation $-2\lambda - x + \lambda x = 0$, we know that the prediction changes and only change when $\lambda = \frac{x}{x-2}$. Let $x = \frac{2\lambda}{\lambda - 1} \geq 0$, then $\hat{y}_3 = -1$ when $\lambda < \lambda'$ and $\hat{y}_3 = 1$ when $\lambda \geq \lambda'$, which completes our proof. $\qquad\square$

The remaining proof is exactly the same as Lemma A.2 and Theorem 3.1, by simply replacing notation $\alpha$ with $\lambda$.

### A.6 PROOF OF THEOREM 3.3

**Upper Bound.** Using Lemma 3.5, we know that each entry of $F^*$ is

$$
F_{ij}^*(\delta) = \frac{1}{\det(I - c \cdot S)} \sum_{k=1}^{n} [\mathrm{adj}(I - c \cdot S)]_{ik} Y_{kj} = \frac{1}{\det(I - c \cdot S)} \sum_{k=1}^{n} (a_{ik} Y_{kj}) \exp(\delta \ln(d_i^{-1} d_k)).
$$

Recall that the prediction on a node is made by $\hat{y}_i = \mathrm{argmax}(F_i^*)$, so the prediction changes only when

$$
F_{ic_1}^* - F_{ic_2}^* = \frac{1}{\det(I - c \cdot S)} \left( \sum_{k=1}^{n} (a_{ik} Y_{kc_1}) \exp(\delta \ln(d_i^{-1} d_k)) - \sum_{k=1}^{n} (a_{ik} Y_{kc_2}) \exp(\delta \ln(d_i^{-1} d_k)) \right)
$$

$$
= \frac{1}{\det(I - c \cdot S)} \left( \sum_{k=1}^{n} (a_{ik} (Y_{kc_1} - Y_{kc_2})) \exp(\delta \ln(d_i^{-1} d_k)) \right)
$$

$$
= 0.
$$

By Lemma 3.6, $F_{ic_1}^* - F_{ic_2}^*$ has at most $n-1$ roots, so the prediction on node $i$ can change at most $n-1$ times. As $\delta$ vary, the prediction can change at most $\binom{c}{2}O(n) \in O(nc^2)$ times. For $n$ nodes and $m$ problem instances, this implies that we have at most $O(mn^2c^2)$ distinct values of loss. The pseudo-dimension $m$ then satisfies $2^m \leq O(mn^2c^2)$, or $m = O(\log nc)$.

**Lower Bound** We construct the small connected component as follows:

**Lemma A.4.** *Consider when $c \geq 1/2$. Given $x \in [\log(2c)/\log(2), 1)$, there exists a labeling instance $(G, L)$ with $4$ nodes, such that the predicted label of the unlabeled points changes only at $\delta = x$ as $\delta$ varies in $(0, 1)$.*

*Proof.* We use binary labeling $a$ and $b$. We have two points labeled $a$ (namely $a_1, a_2$), and one point labeled $b$ (namely $b_1$) connected with both $a_1$ and $a_2$ with edge weight $1$. We also have an unlabeled point $u$ connected to $b_1$ with edge weight $x \geq 0$. That is, the affinity matrix and initial labels are

$$W = \begin{bmatrix} 0 & 1 & 1 & x \\ 1 & 0 & 0 & 0 \\ 1 & 0 & 0 & 0 \\ x & 0 & 0 & 0 \end{bmatrix}, Y = \begin{bmatrix} 1 & 0 \\ 0 & 1 \\ 0 & 1 \\ 0 & 0 \end{bmatrix}.$$

Recall that the score matrix is

$$F^* = (I - c \cdot S)^{-1}Y,$$

where $S = D^{-\delta}WD^{\delta-1}$ and $D$ is diagonal with $D_{ii} = \sum_i W_{ij}$. We now calculate:

$$S = D^{-\delta}WD^{\delta-1} = \begin{bmatrix} 0 & (x+2)^{-\delta} & (x+2)^{-\delta} & x^\delta(x+2)^{-\delta} \\ (x+2)^{-\delta} & 0 & 0 & 0 \\ (x+2)^{-\delta} & 0 & 0 & 0 \\ x^\delta(x+2)^{-\delta} & 0 & 0 & 0 \end{bmatrix},$$

$$\det(I - c \cdot S) = \det \begin{bmatrix} 1 & -c(x+2)^{-\delta} & -c(x+2)^{-\delta} & -cx^\delta(x+2)^{-\delta} \\ -c(x+2)^{-\delta} & 1 & 0 & 0 \\ -c(x+2)^{-\delta} & 0 & 1 & 0 \\ -cx^\delta(x+2)^{-\delta} & 0 & 0 & 1 \end{bmatrix}$$

$$= 1 - c^2 \neq 0,$$

so $(I - c \cdot S)$ is invertible on our instance.

Recall that the prediction on the unlabeled point is $\hat{y}_4 = \arg\max F_4^*$, so we calculate

$$\hat{y}_4 = \text{sign}(F_{4,2}^* - F_{4,1}^*) = \text{sign}\left(\frac{c \cdot x^{1-\delta}(2c - (x+2)^\delta)}{(1-c^2)(x+2)}\right) = \text{sign}\left(2c - (x+2)^\delta\right).$$

$$\text{(since } c \in (0, 1), \text{ and } x \geq 0)$$

Solving the equation $2c - (x+2)^\delta = 0$, we know that the prediction changes and only change when $\delta = \frac{\ln(2c)}{\ln(x+2)}$. Since $x \leq \ln(2c)/\ln(2) \leq 1$, we can let $x = (2c)^{1/x} - 2 \geq 0$, then $\hat{y}_4 = 0$ when $\alpha < x$ and $\hat{y}_4 = 1$ when $\alpha \geq x$, which completes our proof. $\qquad\square$

# B INTRODUCTION TO GAT AND GCN

Here, we provide a brief introduction to GAT and GCN.

**Graph Convolutional Neural Networks (GCNs)** The fundamental idea behind GCNs is to repeatedly apply the convolution operator on graphs (Kipf & Welling, 2017). Define $h_i^0 = z_i$ as the input feature of the $i$-th node and let $h_i^\ell$ be the feature of the $\ell$-th layer of the $i$-th node. We have the following update rule for the features of $h_i^\ell$

$$h_i^\ell = \sigma\left(\sum_{j \in \mathcal{N}_i} \frac{1}{\sqrt{d_id_j}}U^{\ell-1}h_j^{\ell-1}\right)$$

where $d_i$ represents the degree of vertex $i$, $U^\ell$ represents the learnable weights in our model, $\mathcal{N}_i$ represents the neighbors of vertex $i$, and $\sigma(\cdot)$ is the activation function.

**Graph Attention Neural Networks (GATs)**   GAT is a more recent architecture that leverages the self-attention mechanisms to capture the importance of neighboring nodes to generate the features of the next layer (Veličković, Petar et al., 2018). One of the advantages of GAT is its ability to capture long-range dependencies within the graph while giving more weight to influential nodes. This makes GAT particularly effective for tasks involving irregular graph structures and tasks where global context is essential.

Different from GCN, GAT uses the update rule for each layer

$$h_i^\ell = \sigma(\sum_{j \in \mathcal{N}_i} e_{ij}^{\ell-1} U^{\ell-1} h_j^{\ell-1}),$$

where

$$e_{ij}^\ell = \frac{\exp(\hat{e}_{ij}^\ell)}{\sum_{j' \in \mathcal{N}_i} \exp(\hat{e}_{ij'}^\ell)}, \quad \hat{e}_{ij}^\ell = \sigma(V^\ell[U^\ell h_i^\ell, U^\ell h_j^\ell]). \tag{1}$$

Here $e_{ij}^\ell$ is the attention score of node $j$ for node $i$ and $V^\ell$ and $U^\ell$ are learnable parameters.

## C   PROOFS IN SECTION 4

We provide additional proof details from Section 4 below.

### C.1   PROOF OF THEOREM 4.2

**Lemma C.1.** *The $l_2$ norm of different embedding vectors produced by $(\beta, \theta), (\beta', \theta')$ after they process the tree all the way from the leaf level to the root can be bounded as*

$$\Delta_{L,i} \leq \left( \frac{C_{dl}^2 + C_{dh}^2 + C_{dh}}{C_{dl}^3} \right) \|T_{L-1,i}(\beta, \theta)\| \|\beta - \beta'\| + \left( \frac{1}{C_{dl} + 1} + \frac{C_{dh}}{C_{dl}} \right) \Delta_{L-1,i}$$

*Proof.*

$$\begin{aligned}
\Delta_{L,i} =& \|T_{L,i}(\beta, \theta) - T_L(\beta', \theta')\| \\
=& \| \left( \frac{\beta}{d_i + \beta} T_{L-1,i}(\beta, \theta) + \sum_{j=1}^n \frac{w_{ij} T_{L-1,j}(\beta, \theta)}{\sqrt{(d_i + \beta)(d_j + \beta)}} \right) \\
& - \left( \frac{\beta'}{d_i + \beta'} T_{L-1,i}(\beta', \theta') + \sum_{j=1}^n \frac{w_{ij} T_{L-1,j}(\beta', \theta')}{\sqrt{(d_i + \beta')(d_j + \beta')}} \right) \| \\
\leq& \| \left( \frac{\beta}{d_i + \beta} T_{L-1,i}(\beta, \theta) - \frac{\beta'}{d_i + \beta'} T_{L-1,i}(\beta', \theta') \right) \| \\
& + \sum_{j=1}^n \left( \|w_{ij}\| \| \left( \frac{T_{L-1,j}(\beta, \theta)}{\sqrt{(d_i + \beta)(d_j + \beta)}} - \frac{T_{L-1,j}(\beta', \theta')}{\sqrt{(d_i + \beta')(d_j + \beta')}} \right) \| \right)
\end{aligned}$$
$$\text{(by triangle inequality)}$$

The first part can be bounded as

$$\begin{aligned}
& \left\| \frac{\beta}{d_i + \beta} T_{L-1,i}(\beta, \theta) - \frac{\beta'}{d_i + \beta'} T_{L-1,i}(\beta', \theta') \right\| \\
\leq& \left\| \frac{\beta}{d_i + \beta} T_{L-1,i}(\beta, \theta) - \frac{\beta'}{d_i + \beta'} T_{L-1,i}(\beta, \theta) \right\| \\
& + \left\| \frac{\beta'}{d_i + \beta'} T_{L-1,i}(\beta, \theta) - \frac{\beta'}{d_i + \beta'} T_{L-1,i}(\beta', \theta') \right\| \quad \text{(by triangle inequality)} \\
\leq& \left\| \frac{\beta}{d_i + \beta} - \frac{\beta'}{d_i + \beta'} \right\| \|T_{L-1,i}(\beta, \theta)\| + \left\| \frac{\beta'}{d_i + \beta'} \right\| \Delta_{L-1,i} \quad \text{(by Cauchy-Schwarz inequality)}
\end{aligned}$$

Since $\beta \in [0, 1]$ and $d_i \in [C_{dl}, C_{dh}]$, we have

$$\left\|\frac{\beta'}{d_i + \beta'}\right\| = \frac{\beta'}{d_i + \beta'} \le \frac{1}{C_{dl} + 1},$$

and

$$\left\|\frac{\beta}{d_i + \beta} - \frac{\beta'}{d_i + \beta'}\right\| = \left\|\frac{d_i(\beta - \beta')}{(d_i + \beta)(d_i + \beta')}\right\| \le \|\beta - \beta'\|\frac{1}{C_{dl}}.$$

For the second term, let's consider each element in the summation. Using a similar method as above, we get

$$\left\|\frac{T_{L-1,j}(\beta, \theta)}{\sqrt{(d_i + \beta)(d_j + \beta)}} - \frac{T_{L-1,j}(\beta', \theta')}{\sqrt{(d_i + \beta')(d_j + \beta')}}\right\|$$

$$\le \left\|\frac{T_{L-1,j}(\beta, \theta)}{\sqrt{(d_i + \beta)(d_j + \beta)}} - \frac{T_{L-1,j}(\beta, \theta)}{\sqrt{(d_i + \beta')(d_j + \beta')}}\right\|$$

$$+ \left\|\frac{T_{L-1,j}(\beta, \theta)}{\sqrt{(d_i + \beta')(d_j + \beta')}} - \frac{T_{L-1,j}(\beta', \theta')}{\sqrt{(d_i + \beta')(d_j + \beta')}}\right\| \qquad \text{(by triangle inequality)}$$

$$\le \left\|\frac{1}{\sqrt{(d_i + \beta)(d_j + \beta)}} - \frac{1}{\sqrt{(d_i + \beta')(d_j + \beta')}}\right\| \|T_{L-1,j}(\beta, \theta)\|$$

$$+ \left\|\frac{1}{\sqrt{(d_i + \beta')(d_j + \beta')}}\right\| \Delta_{L-1,i} \qquad \text{(Cauchy-Schwarz inequality)}$$

Using the bounds on $\beta$ and $d_i$, we have

$$\left\|\frac{1}{\sqrt{(d_i + \beta')(d_j + \beta')}}\right\| \le \frac{1}{C_{dl}},$$

and

$$\left\|\frac{1}{\sqrt{(d_i + \beta)(d_j + \beta)}} - \frac{1}{\sqrt{(d_i + \beta')(d_j + \beta')}}\right\|$$

$$= \left\|\frac{(d_i + \beta)(d_j + \beta) - (d_i + \beta')(d_j + \beta')}{\sqrt{(d_i + \beta)(d_j + \beta)(d_i + \beta')(d_j + \beta')}[\sqrt{(d_i + \beta)(d_j + \beta)} + \sqrt{(d_i + \beta')(d_j + \beta')}]}\right\|$$

$$\le \left\|\frac{(d_i + d_j + \beta + \beta')(\beta - \beta')}{(C_{dl} + \beta)(C_{dl} + \beta')[(C_{dl} + \beta) + (C_{dl} + \beta')]}\right\|$$

$$\le \frac{C_{dh} + 1}{C_{dl}^3}\|\beta - \beta'\|$$

Combining these results together, we get

$$\Delta_{L,i} \le \frac{1}{C_{dl}}\|\beta - \beta'\|\|T_{L-1,i}(\beta, \theta)\| + \frac{1}{C_{dl} + 1}\Delta_{L-1,i}$$

$$+ \sum_{i=1}^{n}\left(\|w_{ij}\|\left(\frac{(C_{dh} + 1)\|T_{L-1,i}(\beta, \theta)\|}{C_{dl}^3}\|\beta - \beta'\| + \frac{1}{C_{dl}}\Delta_{L-1,i}\right)\right)$$

$$= \frac{1}{C_{dl}}\|\beta - \beta'\|\|T_{L-1,i}(\beta, \theta)\| + \frac{1}{C_{dl} + 1}\Delta_{L-1,i}$$

$$+ d_i\left(\frac{(C_{dh} + 1)\|T_{L-1,i}(\beta, \theta)\|}{C_{dl}^3}\|\beta - \beta'\| + \frac{1}{C_{dl}}\Delta_{L-1,i}\right)$$

$$\le \left(\frac{1}{C_{dl}} + \frac{(C_{dh} + 1)C_{dh}}{C_{dl}^3}\right)\|T_{L-1,i}(\beta, \theta)\|\|\beta - \beta'\| + \left(\frac{1}{C_{dl} + 1} + \frac{C_{dh}}{C_{dl}}\right)\Delta_{L-1,i}$$

$$\le \left(\frac{C_{dl}^2 + C_{dh}^2 + C_{dh}}{C_{dl}^3}\right)\|T_{L-1,i}(\beta, \theta)\|\|\beta - \beta'\| + \left(\frac{1 + C_{dh}}{C_{dl}}\right)\Delta_{L-1,i}$$

$$\qquad\qquad\qquad\qquad\qquad\qquad\qquad\qquad\qquad\qquad\qquad\qquad\qquad\qquad\qquad \square$$

**Lemma C.2.** *The term $\|T_{L-1,i}(\beta, \theta)\|$ satisfies*

$$\|T_{L-1,i}(\beta, \theta)\| \leq \left( \frac{\beta + C_{dh}C_z}{C_{dl} + \beta} \right)^L B_x B_\theta$$

*Proof.*

$$\|T_{L-1,i}(\beta, \theta)\| = \|\frac{\beta}{d_i + \beta} T_{L-2,i}(\beta, \theta) + \sum_{j=1}^{n} \frac{w_{ij} T_{L-2,j}(\beta, \theta)}{\sqrt{(d_i + \beta)(d_j + \beta)}}\|$$

$$\leq \|\frac{\beta}{d_i + \beta} T_{L-2,i}(\beta, \theta)\| + \sum_{j=1}^{n} \|\frac{w_{ij} T_{L-2,j}(\beta, \theta)}{\sqrt{(d_i + \beta)(d_j + \beta)}}\| \quad \text{(by triangle inequality)}$$

$$\leq \frac{\beta}{d_i + \beta} \|T_{L-2,i}(\beta, \theta)\| + \sum_{j=1}^{n} \|w_{ij}\| \|\frac{T_{L-2,j}(\beta, \theta)}{\sqrt{(d_i + \beta)(d_j + \beta)}}\|$$

$$\text{(by Cauchy-Schwarz)}$$

$$\leq \frac{\beta}{d_i + \beta} \|T_{L-2,i}(\beta, \theta)\| + C_{dh} \max_j \|\frac{T_{L-2,j}(\beta, \theta)}{\sqrt{(d_i + \beta)(d_j + \beta)}}\|$$

$$\leq \frac{\beta}{C_{dl} + \beta} \|T_{L-2,i}(\beta, \theta)\| + \frac{C_{dh}}{C_{dl} + \beta} \max_j \|T_{L-2,j}(\beta, \theta)\|$$

$$\leq \left( \frac{C_{dh} + \beta}{C_{dl} + \beta} \right)^{L-1} \|z_i \theta_i\| \quad \text{(by recursively bounding } \|T_{l,i}(\beta, \theta)\|)$$

$$\leq \left( \frac{C_{dh}}{C_{dl}} \right)^{L-1} C_z C_\theta$$

$\square$

**Lemma C.3.** *The change in margin loss for each node, due to change in parameters, after $L$ layers is*

$$\Lambda_i \leq \frac{2}{\gamma} \left( \left( \frac{C_{dl}^2 + C_{dh}^2 + C_{dh}}{C_{dl}^3} \right) \left( \frac{C_{dh}}{C_{dl}} \right)^{L-1} C_z C_\theta \|\beta - \beta'\| \cdot \frac{k_1 - k_1^L}{1 - k_1} + k_1^L C_z \|\theta_i - \theta_i'\| \right),$$

*where $k_1 = (1 + C_{dh}/C_{dl})$.*

*Proof.* From previous lemmas, we know how to recursively bound $\Delta_{L,i}$ using $\Delta_{L-1,i}$, but it remains for us to bound the base case $\Delta_{0,i}$. We have

$$\Delta_{0,i} = \|T_{0,i}(\beta, \theta) - T_{0,i}(\beta, \theta)\| = \|z_i \theta_i - z_i \theta_i'\| \leq \|z_i\| \|\Theta_i - \theta_i'\| \leq C_z \|\theta_i - \theta_i'\|,$$

where the inequality is by Cauchy-Schwarz. For the simplicity of notation, let $\bar{T}_L$ be the bound we derived for $\|T_{L-1,i}(\beta, \theta)\|$ from the previous lemma. We have

$$\Delta_{L,i} \leq \left( \frac{C_{dl}^2 + C_{dh}^2 + C_{dh}}{C_{dl}^3} \right) \|T_{L-1,i}(\beta, \theta)\| \|\beta - \beta'\| + \left( \frac{1 + C_{dh}}{C_{dl}} \right) \Delta_{L-1,i}$$

$$\leq \left( \frac{C_{dl}^2 + C_{dh}^2 + C_{dh}}{C_{dl}^3} \right) \bar{T}_L \|\beta - \beta'\| + \left( \frac{1 + C_{dh}}{C_{dl}} \right) \Delta_{L-1,i}$$

$$= \left( \frac{C_{dl}^2 + C_{dh}^2 + C_{dh}}{C_{dl}^3} \right) \bar{T}_L \|\beta - \beta'\| \cdot \sum_{l=0}^{L-1} \left( \frac{1 + C_{dh}}{C_{dl}} \right)^l + \left( \frac{1 + C_{dh}}{C_{dl}} \right)^L \cdot \Delta_{0,i}$$

$$\text{(by recursively bounding the terms)}$$

$$= \left( \frac{C_{dl}^2 + C_{dh}^2 + C_{dh}}{C_{dl}^3} \right) \bar{T}_L \|\beta - \beta'\| \cdot \frac{k_1 - k_1^L}{1 - k_1} + k_1^L C_z \|\theta_i - \theta_i'\|$$

where

$$k_1 = \frac{1 + C_{dh}}{C_{dl}}.$$

The change in margin loss for each node after $L$ layers is then

$$
\begin{aligned}
\Lambda_i &= |g_\gamma(-\tau(f_{\beta,\theta}(x_i), y_i)) - g_\gamma(-\tau(f_{\beta',\theta'}(x_i), y_i))| \\
&\leq \frac{1}{\gamma} |\tau(f_{\beta,\theta}(x_i), y_i)) - \tau(f_{\beta',\theta'}(x_i), y_i))| && \text{(since } g_\gamma \text{ is } 1/\gamma\text{-Lipschitz)} \\
&= \frac{1}{\gamma} |(2f_{\beta,\theta}(x_i) - 1)y_i - (2f_{\beta',\theta'}(x_i) - 1)y_i| \\
&\leq \frac{2}{\gamma} |y_i| |f_{\beta,\theta}(x_i) - f_{\beta',\theta'}(x_i)| && \text{(by Cauchy-Schwarz inequality)} \\
&\leq \frac{2}{\gamma} |\sigma(T_{L,i}(\beta,\theta)) - \sigma(T_{L,i}(\beta',\theta'))| && \text{(since } y_i \in \{-1,1\}) \\
&\leq \frac{2}{\gamma} |T_{L,i}(\beta,\theta) - T_{L,i}(\beta',\theta')| && \text{(since sigmoid is 1-Lipschitz)} \\
&= \frac{2}{\gamma} \Delta_{L,i} \\
&\leq \frac{2}{\gamma} \left( \left( \frac{C_{dl}^2 + C_{dh}^2 + C_{dh}}{C_{dl}^3} \right) \left( \frac{C_{dh}}{C_{dl}} \right)^{L-1} C_z C_\theta \left( \frac{k_1 - k_1^L}{1 - k_1} \right) \|\beta - \beta'\| + k_1^L C_z \|\theta_i - \theta_i'\| \right)
\end{aligned}
$$

$\square$

**Lemma C.4.** *The change in margin loss $\Lambda_i$ for each node can be bounded by $\epsilon$, using a covering of size P, where P depends on $\epsilon$.*

*Proof.* Let $k_2 = \frac{2}{\gamma} \left( \frac{C_{dl}^2 + C_{dh}^2 + C_{dh}}{C_{dl}^3} \right) \left( \frac{C_{dh}}{C_{dl}} \right)^{L-1} C_z C_\theta \left( \frac{k_1 - k_1^L}{1 - k_1} \right)$ and $k_3 = \frac{2}{\gamma} k_1^L C_z$ for simplicity of notation.

We begin by noting that we can find a covering $\mathcal{C}\left(\beta, \frac{\epsilon}{4k_2}, |\cdot|\right)$ of size

$$
\mathcal{N}\left(\beta, \frac{\epsilon}{4k_2}, |\cdot|\right) \leq \frac{8k_2}{\epsilon} + 1.
$$

Also, we can find a covering $\mathcal{C}\left(\theta, \frac{\epsilon}{4k_3}, \|\cdot\|\right)$ of size

$$
\mathcal{N}\left(\theta, \frac{\epsilon}{4k_3}, \|\cdot\|\right) \leq \left(\frac{8k_3}{\epsilon} + 1\right)^d.
$$

Thus, for any specified $\epsilon$, we can ensure that $\Lambda_i$ is at most $\epsilon$ with a covering number

$$
P \leq \mathcal{N}\left(\beta, \frac{\epsilon}{4k_2}, |\cdot|\right) \mathcal{N}\left(\Theta, \frac{\epsilon}{4k_3}, \|\cdot\|\right) \leq \left(\frac{8\max\{k_2, k_3\}}{\epsilon} + 1\right)^{d+1}.
$$

When $\epsilon < 8\max\{k_2, k_3\}$, we have

$$
\log P \leq (d+1) \log\left(\frac{16\max\{k_2, k_3\}}{\epsilon}\right).
$$

$\square$

We can now finish our proof for Lemma 4.2.

*Proof.* Using Lemma A.5 from Bartlett et al. (2017), we obtain that

$$
\hat{R}_{\mathcal{T}}(\mathcal{H}_{(\beta,\theta)}^\gamma) \leq \inf_{\alpha>0}\left(\frac{4\alpha}{\sqrt{m}} + \frac{12}{m} \int_\alpha^{\sqrt{m}} \sqrt{\log \mathcal{N}(\mathcal{H}_{(\beta,\theta)}^\gamma, \epsilon, \|\cdot\|)} d\epsilon\right).
$$

Using the previous lemmas, we have

$$\int_\alpha^{\sqrt{m}} \sqrt{\log \mathcal{N}(\mathcal{H}_{(\beta,\theta)}^\gamma, \epsilon, \|\cdot\|)} d\epsilon = \int_\alpha^{\sqrt{m}} \sqrt{\log P} d\epsilon$$

$$\leq \int_\alpha^{\sqrt{m}} \sqrt{(d+1) \log \left( \frac{16 \max\{k_2, k_3\}}{\epsilon} \right)} d\epsilon$$

$$\leq \sqrt{m} \sqrt{(d+1) \log \left( \frac{16 \max\{k_2, k_3\}}{\alpha} \right)}$$

Plugging in $\alpha = \sqrt{\frac{1}{m}}$, we have

$$\hat{R}_{\mathcal{T}}(\mathcal{H}_{(\beta,\theta)}^\gamma) \leq \frac{4}{m} + \frac{12\sqrt{(d+1) \log(16\sqrt{m} \max\{k_2, k_3\})}}{\sqrt{m}}.$$

$\square$

## C.2 PROOF OF THEOREM 4.3

**Lemma C.5.** *For any $z, z', \in \mathbb{R}^{d \times r}$ and $b, b' \in \mathbb{R}^{r \times t}$ such that $\|z\|_F \leq C_z, \|z'\|_F \leq C_z, \|b\|_F \leq C_b, \|b'\|_F \leq C_b$, we have*

$$\|zb - z'b'\|_F \leq C_z \|b - b'\|_F + C_b \|z - z'\|_F.$$

*The result also holds when $z, b, z', b'$ are vectors or real numbers. The corresponding norms are $\|\cdot\|$ and $|\cdot|$.*

*Also, by recursively using the inequality above, we may have that for any $z_1, \ldots, z_n$ and $z'_1, \ldots, z'_n$ such that $\|z_i\| \leq C_i, \|z'_i\| \leq C_i$,*

$$\|z_1 z_2 \ldots z_n - z'_1 z'_2 \ldots z'_n\| \leq \sum_{i=1}^{n} \left( \|z_i - z'_i\| \prod_{j \in [n], j \neq i} C_j \right).$$

*Here, for simplicity of notation, we used $\|\cdot\|$ to denote the type of norm that corresponds to the dimension of the $z_i$'s.*

*Proof.*

$$\|ab - a'b'\|_F = \|ab - a'b' + ab' - ab'\|_F$$
$$\leq \|ab - ab'\|_F + \|ab' - a'b'\|_F \qquad \text{(by triangle inequality)}$$
$$\leq \|a\|_F \|b - b'\|_F + \|b'\|_F \|a - a'\|_F \qquad \text{(by Cauchy-Schwarz inequality)}$$
$$\leq C_z \|b - b'\|_F + C_b \|a - a'\|_F$$

$\square$

**Lemma C.6.** *The $l_2$ norm of different embedding vectors at level $L$, $h_i^L$, produced by $(\alpha, U, V), (\alpha', U', V')$ after they process the tree all the way from the leaf level to the root can be bounded as*

$$\Delta_{i,L} \leq C_U (\max_{j \in \mathcal{N}_i} \|h_j^{L-1}\|) |\eta - \eta'| + r C_U (\max_{j \in \mathcal{N}_i} \|h_j^{L-1}\|) + (\max_{j \in \mathcal{N}_i} \|h_j^{L-1}\|) \|U - U'\|$$

$$+ C_U (\max_{j \in \mathcal{N}_i} \|h_j^{L-1} - h_j'^{(L-1)}\|) + \frac{2r C_U}{C_{dl}} \|h_i^{L-1} - h_i'^{(L-1)}\|$$

$$+ \frac{2r}{C_{dl}} \|h_i^{L-1}\| \|U - U'\| + \frac{2r C_U}{C_{dl}} |\eta - \eta'|$$

*Proof.*

$$\Delta_{i,L} = \|h_i^L(\eta, U, V) - h_i^L(\eta', U', V')\|$$

$$= \left\| \sigma \left( \sum_{j \in \mathcal{N}_i} \left( \eta \cdot e_{ij}^{L-1} + (1-\eta) \cdot \frac{1}{\sqrt{d_i d_j}} \right) U h_j^{L-1} \right) \right.$$

$$\left. - \sigma \left( \sum_{j \in \mathcal{N}_i} \left( \eta' \cdot e_{ij}'^{(L-1)} + (1-\eta') \cdot \frac{1}{\sqrt{d_i d_j}} \right) U' h_j'^{(L-1)} \right) \right\|$$

$$\leq \left\| \sum_{j \in \mathcal{N}_i} \left( (\eta \cdot e_{ij}^{L-1} U h_j^{L-1}) - (\eta' \cdot e_{ij}'^{(L-1)} U' h_j'^{(L-1)}) \right) \right.$$

$$\left. + \sum_{j \in \mathcal{N}_i} \left( (1-\eta) \cdot \frac{1}{\sqrt{d_i d_j}} U h_j^{L-1} - (1-\eta') \cdot \frac{1}{\sqrt{d_i d_j}} U' h_j'^{(L-1)} \right) \right\|$$

$$\text{(since } \sigma \text{ is 1-Lipschitz)}$$

$$\leq \sum_{j \in \mathcal{N}_i} \| (\eta \cdot e_{ij}^{L-1} U h_j^{L-1}) - (\eta' \cdot e_{ij}'^{(L-1)} U' h_j'^{(L-1)}) \|$$

$$+ \sum_{j \in \mathcal{N}_i} \| (1-\eta) \cdot \frac{1}{\sqrt{d_i d_j}} U h_i^{L-1} - (1-\eta') \cdot \frac{1}{\sqrt{d_i d_j}} U' h_i'^{(L-1)} \| \quad \text{(by triangle inequality)}$$

Using Lemma C.5, we can bound each term in the first summation as

$$\| (\eta \cdot e_{ij}^{L-1} U h_j^{L-1}) - (\eta' \cdot e_{ij}'^{(L-1)} U' h_j'^{(L-1)}) \|$$

$$\leq C_U \bar{e}_{ij}^{L-1} \bar{h}_j^{L-1} \cdot |\eta - \eta'| + C_U \bar{h}_j^{L-1} \cdot |e_{ij}^{L-1} - e_{ij}'^{(L-1)}|$$

$$+ \bar{e}_{ij}^{L-1} \bar{h}_j^{L-1} \| U - U' \| + C_U \bar{e}_{ij}^{L-1} \| h_j^{L-1} - h_j'^{(L-1)} \|$$

Here, $\bar{h}_j^{L-1}$ is an upper bound on $\|h_j^{L-1}\|$ and $\|h_j'^{(L-1)}\|$, and $\bar{e}_{ij}^{L-1}$ is an upper bound on $|e_{ij}^{L-1}|$ and $|e_{ij}'^{(L-1)}|$.

Bounding each term in the second summation, we have

$$\| (1-\eta) \cdot \frac{1}{\sqrt{d_i d_j}} U h_i^{L-1} - (1-\eta') \cdot \frac{1}{\sqrt{d_i d_j}} U' h_i'^{(L-1)} \|$$

$$\leq \| \frac{1}{\sqrt{d_i d_j}} U h_i^{L-1} - \frac{1}{\sqrt{d_i d_j}} U' h_i'^{(L-1)} \| + \| \eta \cdot \frac{1}{\sqrt{d_i d_j}} U h_i^{L-1} - \eta' \cdot \frac{1}{\sqrt{d_i d_j}} U' h_i'^{(L-1)} \|$$

$$\text{(by triangle inequality)}$$

$$\leq \frac{1}{C_{dl}} \| U h_i^{L-1} - U' h_i'^{(L-1)} \| + \frac{1}{C_{dl}} \| \eta \cdot U h_i^{L-1} - \eta' \cdot U' h_i'^{(L-1)} \|$$

$$\leq \frac{1}{C_{dl}} \left( C_U \| h_i^{L-1} - h_i'^{(L-1)} \| + \bar{h}_i^{L-1} \| U - U' \| \right)$$

$$+ \frac{1}{C_{dl}} \left( C_U \| h_i^{L-1} - h_i'^{(L-1)} \| + \bar{h}_i^{L-1} \| U - U' \| + C_U \bar{h}_i^{L-1} |\eta - \eta'| \right) \quad \text{(using Lemma C.5)}$$

$$= \frac{1}{C_{dl}} \left( 2 C_U \| h_i^{L-1} - h_i'^{(L-1)} \| + 2 \bar{h}_i^{L-1} \| U - U' \| + C_U \bar{h}_i^{L-1} |\eta - \eta'| \right).$$

Combining the above results, we have

$$
\begin{aligned}
\Delta_i^L \leq &\sum_{j \in \mathcal{N}_i} \Big( C_U \bar{e}_{ij}^{L-1} \bar{h}_j^{L-1} \cdot |\eta - \eta'| + C_U \bar{h}_j^{L-1} \cdot |e_{ij}^{L-1} - e_{ij}'^{(L-1)}| \\
&+ \bar{e}_{ij}^{L-1} \bar{h}_j^{L-1} \|U - U'\| + C_U \bar{e}_{ij}^{L-1} \|h_j^{L-1} - h_j'^{(L-1)}\| \\
&+ \frac{1}{C_{dl}} \big( 2C_U \|h_i^{L-1} - h_i'^{(L-1)}\| + 2\bar{h}_i^{L-1} \|U - U'\| + C_U \bar{h}_i^{L-1} |\eta - \eta'| \big) \Big) \\
\leq &C_U (\max_{j \in \mathcal{N}_i} \bar{h}_j^{L-1}) |\eta - \eta'| + rC_U (\max_{j \in \mathcal{N}_i} \bar{h}_j^{L-1}) + (\max_{j \in \mathcal{N}_i} \bar{h}_j^{L-1}) \|U - U'\| \\
&+ C_U (\max_{j \in \mathcal{N}_i} \|h_j^{L-1} - h_j'^{(L-1)}\|) + \frac{2rC_U}{C_{dl}} \|h_i^{L-1} - h_i'^{(L-1)}\| \\
&+ \frac{2r}{C_{dl}} \bar{h}_i^{L-1} \|U - U'\| + \frac{2rC_U}{C_{dl}} |\eta - \eta'| \\
&\quad\quad\quad\quad (\text{since } e_{ij}^\ell \leq 1, \sum_{j \in \mathcal{N}_i} e_{ij}^\ell = 1, \text{ and the branching factor is } r)
\end{aligned}
$$

It remains for us to derive $\bar{h}_j^{L-1}$ for all $j$. $\qquad\qquad\qquad\qquad\qquad\qquad\qquad\qquad \square$

**Lemma C.7.** *We can upper bound the norm of node feature embedding at level $\ell + 1$ by*

$$
\|h_i^\ell\| \leq r^\ell C_U^{\ell+1} C_z \max(1, \frac{1}{C_{dl}})^\ell.
$$

*Proof.*

$$
\begin{aligned}
\|h_i^{\ell+1}\| &= \|\sigma \left( \sum_{j \in \mathcal{N}_i} (\eta \cdot e_{ij}^\ell + (1-\eta) \cdot \frac{1}{\sqrt{d_i d_j}}) U h_j^\ell \right) \| \\
&\leq \| \sum_{j \in \mathcal{N}_i} (\eta \cdot e_{ij}^\ell + (1-\eta) \cdot \frac{1}{\sqrt{d_i d_j}}) U h_j^\ell \| \quad\quad (\text{since } \|\sigma(x)\| \leq \|x\|) \\
&\leq \sum_{j \in \mathcal{N}_i} \left| \eta \cdot e_{ij}^\ell + (1-\eta) \cdot \frac{1}{\sqrt{d_i d_j}} \right| \|U\| \|h_j^\ell\| \\
&\quad\quad\quad (\text{by triangle inequality and Cauchy-Schwarz inequality}) \\
&\leq C_U \sum_{j \in \mathcal{N}_i} \left| \eta \cdot e_{ij}^\ell + (1-\eta) \cdot \frac{1}{\sqrt{d_i d_j}} \right| \|h_j^\ell\| \\
&\leq rC_U \max(1, \frac{1}{C_{dl}}) (\max_{j \in \mathcal{N}_i} \|h_j^{\ell-1}\|)
\end{aligned}
$$

Recursively bounding the terms, we have

$$
\|h_i^\ell\| \leq r^\ell C_U^\ell \max(1, \frac{1}{C_{dl}})^\ell \max_{j \in [n]} \|h_j^0\| \leq r^\ell C_U^{\ell+1} C_z \max(1, \frac{1}{C_{dl}})^\ell.
$$

$\qquad\qquad\qquad\qquad\qquad\qquad\qquad\qquad\qquad\qquad\qquad\qquad\qquad\qquad\qquad \square$

**Lemma C.8.** *The change in margin loss due to the change in parameter values after $L$ layers satisfies*

$$
\Lambda_i \leq \frac{2}{k} (k_1 + k_2 |\eta - \eta'| + k_3 \|U - U'\|) \frac{k_4^L - k_4}{k_4 - 1} + k_4 C_z \|U - U'\|,
$$

*where*

$$k_1 = r^L C_U^{L+1} C_z \max(1, \frac{1}{C_{dl}})^{L-1}$$

$$k_2 = r^{L-1} C_U^{L+1} C_z \max(1, \frac{1}{C_{dl}})^{L-1} + \frac{2rC_U}{C_{dl}}$$

$$k_3 = \left(1 + \frac{2r}{C_{dl}}\right) r^{L-1} C_U^L C_z \max(1, \frac{1}{C_{dl}})^{L-1}$$

$$k_4 = C_U + \frac{2rC_U}{C_{dl}}.$$

*Proof.* Using the previous two lemmas, we know

$$\|h_i^L(\eta, U, V) - h_i^L(\eta', U', V')\|$$

$$\leq C_U (\max_{j \in \mathcal{N}_i} \bar{h}_j^{L-1}) |\eta - \eta'| + rC_U (\max_{j \in \mathcal{N}_i} \bar{h}_j^{L-1}) + (\max_{j \in \mathcal{N}_i} \bar{h}_j^{L-1}) \|U - U'\|$$

$$+ C_U (\max_{j \in \mathcal{N}_i} \|h_j^{L-1} - h_j'^{(L-1)}\|) + \frac{2rC_U}{C_{dl}} \|h_i^{L-1} - h_i'^{(L-1)}\| + \frac{2r}{C_{dl}} \bar{h}_i^{L-1} \|U - U'\| + \frac{2rC_U}{C_{dl}} |\eta - \eta'|$$

$$\leq k_1 + k_2 |\eta - \eta'| + k_3 \|U - U'\| + k_4 (\max_{j \in [n]} \|h_j^{L-1} - h_j'^{(L-1)}\|)$$

$$= (k_1 + k_2 |\eta - \eta'| + k_3 \|U - U'\|) \frac{k_4^L - k_4}{k_4 - 1} + k_4 (\max_{j \in [n]} \|h_j^0 - h_j'^0\|)$$

$$\leq (k_1 + k_2 |\eta - \eta'| + k_3 \|U - U'\|) \frac{k_4^L - k_4}{k_4 - 1} + k_4 C_z \|U - U'\|$$

where

$$k_1 = r^L C_U^{L+1} C_z \max(1, \frac{1}{C_{dl}})^{L-1}$$

$$k_2 = r^{L-1} C_U^{L+1} C_z \max(1, \frac{1}{C_{dl}})^{L-1} + \frac{2rC_U}{C_{dl}}$$

$$k_3 = \left(1 + \frac{2r}{C_{dl}}\right) r^{L-1} C_U^L C_z \max(1, \frac{1}{C_{dl}})^{L-1}$$

$$k_4 = C_U + \frac{2rC_U}{C_{dl}}.$$

The change in margin loss for each node after L layers is then

$$\Lambda_i = |g_\gamma(-\tau(f_{\eta, U, V}(x_i), y_i)) - g_\gamma(-\tau(f_{\eta', U', V'}(x_i), y_i))|$$

$$\leq \frac{1}{\gamma} |\tau(f_{\eta, U, V}(x_i), y_i)) - \tau(f_{\eta', U', V'}(x_i), y_i))| \qquad \text{(since } g_\gamma \text{ is } 1/\gamma\text{-Lipschitz)}$$

$$= \frac{1}{\gamma} |(2f_{\beta, \theta}(x_i) - 1)y_i - (2f_{\beta', \theta'}(x_i) - 1)y_i|$$

$$\leq \frac{2}{\gamma} |y_i| |f_{\eta, U, V}(x_i) - f_{\eta', U', V'}(x_i)| \qquad \text{(by Cauchy-Schwarz inequality)}$$

$$\leq \frac{2}{\gamma} |\sigma(h_i^L(\eta, U, V)[0]) - \sigma(h_i^L(\eta', U', V')[0])| \qquad \text{(since } y_i \in \{-1, 1\})$$

$$\leq \frac{2}{\gamma} |h_i^L(\eta, U, V)[0] - h_i^L(\eta', U', V')[0]| \qquad \text{(since } \sigma \text{ is 1-Lipschitz)}$$

$$\leq \frac{2}{\gamma} (k_1 + k_2 |\eta - \eta'| + k_3 \|U - U'\|) \frac{k_4^L - k_4}{k_4 - 1} + k_4 C_z \|U - U'\|.$$

$\square$

**Lemma C.9.** *The change in margin loss $\Lambda_i$ for each node can be bounded by $\epsilon$, using a covering of size $P$, where $P$ depends on $\epsilon$, with*

$$\log P \le (d^2 + 1) \log \left( \frac{8 \max\{A, BC_U \sqrt{d}\}}{\epsilon} \right).$$

*Proof.* We let $A = \frac{2k_2(k_4^L - k_4)}{k(k_4 - 1)}$ and $B = \frac{2k_3(k_4^L - k_4) + \gamma(k_4^2 - k_4)C_z}{\gamma(k_4 - 1)}$ for simplicity of notation. Note that we have $\Lambda_i \le A|\eta - \eta'| + B\|U - U'\|$.

We begin by noting that we can find a covering $\mathcal{C}(\eta, \frac{\epsilon}{2A}, |\cdot|)$ of size

$$\mathcal{N}(\eta, \frac{\epsilon}{2A}, |\cdot|) \le 1 + \frac{4A}{\epsilon}.$$

We can also find a covering $\mathcal{C}(U, \frac{\epsilon}{2B}, \|\cdot\|_F)$ with size

$$\mathcal{N}(U, \frac{\epsilon}{2B}, \|\cdot\|_F) \le \left( 1 + \frac{4BC_U\sqrt{d}}{\epsilon} \right)^{d^2}.$$

For any specified $\epsilon$, we can ensure that $\Lambda_i$ is at most $\epsilon$ with a covering number of

$$P \le \mathcal{N}(\eta, \frac{\epsilon}{2A}, |\cdot|) \cdot \mathcal{N}(U, \frac{\epsilon}{2B}, \|\cdot\|_F)$$

$$\le \left( 1 + \frac{4A}{\epsilon} \right)\left( 1 + \frac{4BC_U\sqrt{d}}{\epsilon} \right)^{d^2} \le (1 + \frac{4\max\{A, BC_U\sqrt{d}\}}{\epsilon})^{d^2+1}$$

Moreover, when $\epsilon \le 4 \max\{A, BC_U\sqrt{d}\}$, we have

$$\log P \le (d^2 + 1) \log \left( \frac{8 \max\{A, BC_U\sqrt{d}\}}{\epsilon} \right).$$

$\square$

Now we can finish our proof for Theorem 4.3.

*Proof.* Using Lemma A.5 from Bartlett et al. (2017), we obtain that

$$\hat{R}_{\mathcal{T}}(\mathcal{H}^\gamma_{(\eta,U,V))}) \le \inf_{\alpha > 0} \left( \frac{4\alpha}{\sqrt{m}} + \frac{12}{m} \int_\alpha^{\sqrt{m}} \sqrt{\log \mathcal{N}(\mathcal{H}^\gamma_{(\eta,U,V))}, \epsilon, \|\cdot\|)} d\epsilon \right).$$

Using the previous lemmas, we have

$$\int_\alpha^{\sqrt{m}} \sqrt{\log \mathcal{N}(\mathcal{H}^\gamma_{(\eta,U,V))}, \epsilon, \|\cdot\|)} d\epsilon = \int_\alpha^{\sqrt{m}} \sqrt{\log P} d\epsilon$$

$$\le \int_\alpha^{\sqrt{m}} \sqrt{(d^2 + 1) \log \left( \frac{8 \max\{A, BC_U\sqrt{d}\}}{\epsilon} \right)} d\epsilon$$

$$\le \sqrt{m} \sqrt{(d^2 + 1) \log \left( \frac{8 \max\{A, BC_U\sqrt{d}\}}{\alpha} \right)}$$

Plugging in $\alpha = \sqrt{\frac{1}{m}}$, we have

$$\hat{R}_{\mathcal{T}}(\mathcal{H}^\gamma_{(\eta,U,V))}) \le \frac{4}{m} + \frac{12\sqrt{(d^2 + 1) \log \left( 8\sqrt{m} \max\{A, BC_U\sqrt{d}\} \right)}}{\sqrt{m}}.$$

$\square$

# D    EXPERIMENTS

In this section, we empirically evaluate our proposed GCAN interpolation methods on nine standard benchmark datasets. Our goal is to see whether tuning $\eta$ gives better results than both GCN and GAT. The setup details of our experiment are described in Appendix D.1.

| Dataset | 0.0 | 0.1 | 0.2 | 0.3 | 0.4 | 0.5 | 0.6 | 0.7 | 0.8 | 0.9 | 1.0 | Rel. GCN | Rel. GAT |
|---|---|---|---|---|---|---|---|---|---|---|---|---|---|
| CIFAR10 | 0.7888 ± 0.0010 | 0.7908 ± 0.0008 | 0.7908 ± 0.0015 | 0.7907 ± 0.0012 | 0.7943 ± 0.0022 | 0.7918 ± 0.0018 | 0.7975 ± 0.0017 | 0.7971 ± 0.0023 | 0.7921 ± 0.0023 | 0.7986 ± 0.0028 | **0.7984 ± 0.0023** | 4.54% | 0% |
| WikiCS | 0.9525 ± 0.0007 | 0.9516 ± 0.0006 | 0.9532 ± 0.0011 | 0.9545 ± 0.0008 | 0.9551 ± 0.0015 | 0.9545 ± 0.0012 | 0.9539 ± 0.0012 | **0.9553 ± 0.0012** | 0.9530 ± 0.0007 | 0.9536 ± 0.0009 | 0.9539 ± 0.0009 | 5.89% | 3.04% |
| Cora | 0.6132 ± 0.0218 | 0.8703 ± 0.0251 | 0.8879 ± 0.0206 | 0.8396 ± 0.0307 | 0.8022 ± 0.0385 | 0.8615 ± 0.0402 | **0.9011 ± 0.0421** | 0.8088 ± 0.0362 | 0.8505 ± 0.0240 | 0.8549 ± 0.0389 | 0.8725 ± 0.0334 | 74.43% | 22.43% |
| Citeseer | **0.7632 ± 0.0052** | 0.6944 ± 0.0454 | 0.7602 ± 0.0566 | 0.7500 ± 0.0461 | 0.7339 ± 0.0520 | 0.7427 ± 0.0462 | 0.7588 ± 0.0504 | 0.7193 ± 0.0567 | 0.7661 ± 0.0482 | 0.7266 ± 0.0412 | 0.7471 ± 0.0444 | 0% | 6.37% |
| PubMed | 0.9350 ± 0.0009 | 0.9306 ± 0.0006 | 0.9356 ± 0.0009 | 0.9281 ± 0.0007 | **0.9356 ± 0.0007** | 0.9319 ± 0.0009 | 0.9313 ± 0.0007 | 0.9288 ± 0.0009 | 0.9313 ± 0.0006 | 0.9338 ± 0.0010 | 0.9356 ± 0.0009 | 0.92% | 0% |
| CoauthorCS | 0.9733 ± 0.0007 | 0.9733 ± 0.0008 | **0.9765 ± 0.0005** | 0.9744 ± 0.0005 | 0.9733 ± 0.0009 | 0.9690 ± 0.0007 | 0.9712 ± 0.0009 | 0.9722 ± 0.0005 | 0.9722 ± 0.0011 | 0.9722 ± 0.0007 | 0.9744 ± 0.0007 | 11.99% | 08.20% |
| AmazonPhotos | 0.9605 ± 0.0022 | 0.9617 ± 0.0007 | 0.9629 ± 0.0015 | 0.9599 ± 0.0013 | 0.9641 ± 0.0017 | 0.9574 ± 0.0018 | 0.9641 ± 0.0019 | 0.9592 ± 0.0133 | **0.9653 ± 0.0027** | 0.9635 ± 0.0031 | 0.9562 ± 0.0019 | 12.15% | 20.77% |
| Actor | 0.5982 ± 0.0016 | 0.5919 ± 0.0022 | **0.6005 ± 0.0039** | 0.5959 ± 0.0039 | 0.5965 ± 0.0027 | 0.5970 ± 0.0037 | 0.5976 ± 0.0037 | 0.5993 ± 0.0043 | 0.5930 ± 0.0037 | 0.5970 ± 0.0037 | 0.5953 ± 0.0031 | 0.57% | 1.28% |
| Cornell | 0.7341 ± 0.0097 | 0.7364 ± 0.0165 | 0.7364 ± 0.0073 | 0.7205 ± 0.0109 | 0.7523 ± 0.0120 | 0.7795 ± 0.0188 | 0.7568 ± 0.0140 | 0.7500 ± 0.0120 | 0.7477 ± 0.0138 | 0.7909 ± 0.0136 | **0.8000 ± 0.0423** | 24.78% | 0% |
| Wisconsin | 0.8688 ± 0.0077 | **0.8922 ± 0.0035** | 0.8688 ± 0.0080 | 0.8906 ± 0.0049 | 0.8797 ± 0.0044 | 0.8578 ± 0.0120 | 0.8875 ± 0.0037 | 0.8781 ± 0.0082 | 0.8563 ± 0.0128 | 0.8750 ± 0.0121 | 0.8719 ± 0.0076 | 17.84% | 15.84% |

Table 2: Results on the proposed GCAN interpolation. Each column corresponds to one $\eta$ value. Each row corresponds to one dataset. Each entry shows the accuracy and the interval. The accuracy with optimal $\eta$ value outperforms both pure GCN and pure GAT. The right two columns show the percentage of prediction error reduction relative to GCN and GAT.

In Table 2, we show the mean accuracy across 30 runs of each $\eta$ value and the $90\%$ confidence interval associated with each experiment. It is interesting to note that for various datasets we see varying optimal $\eta$ values for best performance. More often than not, the best model is interpolated between GCN and GAT, showing that we can achieve an improvement on both baselines simply by interpolating between the two. For example, GCN achieves the best accuracy among all interpolations in Citeseer, but in other datasets such as CIFAR 10 or Wisconsin, we see higher final accuracies when the $\eta$ parameter is closer to $1.0$ (more like GAT). The interpolation between the two points also does not increase or decrease monotonically for many of the datasets. The optimal $\eta$ value for each dataset can be any value between $0.0$ and $1.0$. This suggests that one should be able to learn the best $\eta$ parameter for each specific dataset. By learning the optimal $\eta$ value, we can outperform both GAT and GCN.

## D.1    EXPERIMENT SETUP FOR GCAN

We apply dropout with a probability of $0.4$ for all learnable parameters, apply 1 head of the specialized attention layer (with new update rule), and then an out attention layer. The activation we choose is eLU activation (following prior work Veličković, Petar et al. (2018)), with 8 hidden units, and 3 attention heads.

These GCAN interpolation experiments are all run with only $20\%$ of the dataset being labeled datapoints, and the remaining $80\%$ representing the unlabeled datapoints that we test our classification accuracy on. Table 3 notes the exact setup of each dataset, and the overall training time of each experiment.

| Dataset | Num of train nodes | learn rate | Epoch | Num of exp | Train time(sec) | Dim of hid. layers | Num of Attention Heads |
|---|---|---|---|---|---|---|---|
| CiFAR10 | 400 | 7e-3 | 1000 | 30 | 13.5354 | 1 | 3 |
| WikiCS | 192 | 7e-3 | 1000 | 30 | 6.4742 | 1 | 3 |
| Cora | 170 | 7e-3 | 1000 | 30 | 7.4527 | 1 | 3 |
| Citeseer | 400 | 7e-3 | 1000 | 30 | 6.4957 | 1 | 3 |
| Pubmed | 400 | 7e-3 | 1000 | 30 | 13.1791 | 1 | 3 |
| CoAuthor CS | 400 | 0.01 | 1000 | 30 | 6.8015 | 1 | 3 |
| Amazon Photos | 411 | 0.01 | 400 | 30 | 11.0201 | 1 | 3 |
| Actor | 438 | 0.01 | 1000 | 30 | 14.7753 | 1 | 3 |
| Cornell | 10 | 0.01 | 1000 | 30 | 6.9423 | 1 | 3 |
| Wisconsin | 16 | 0.01 | 1000 | 30 | 6.9271 | 1 | 3 |

Table 3: Experiment setup

For datasets that are not inherently graph-structured (e.g., CIFAR-10), we first compute the Euclidean distance between the feature vectors of each pair of nodes. An edge is then added between two nodes if their distance is below a predefined threshold.

