# OpenReview forum: "Theoretical Analyses of Hyperparameter Selection in Graph-Based Semi-Supervised Learning"
_ICLR.cc/2025/Conference — Submitted to ICLR 2025_

### Official Review · Reviewer_WNgy · 2024-10-26

**Soundness:** 3
**Presentation:** 2
**Contribution:** 2
**Rating:** 6
**Confidence:** 4

**Summary:**

This paper studies the problem of hyperparameter selection in graph-based semi-supervised learning. The theoretic analysis starts from three classical label propagation algorithms: local and global consistency algorithm, smoothing-based algorithm and normalized adjacency matrix-based algorithms. For each of them, the authors show that the upper and lower bound of the pseudo-dimension is of order $\log (n)$, where $n$ is the number of nodes in graph. Then the authors turn to some modern GNN models including SGC, GCN and GAT. Concretely, the authors propose a GNN model named GCAN that linearly combines the update of GCN and GAT via a hyperparameter $\eta$. For the case that $\eta=0$ and $\eta=1$, the GCAN degenerates to GCN and GAT, respectively. For SGC and GCAN, the authors analyze the upper bounds for the Rademacher complexity of tuning the interpolation coefficient. Both of them are of order $\sqrt{\log m/m}$, where $m$ is the number of training nodes with labels. Finally, the authors conduct experiments to demonstrate that GCAN has a matched or better performance compared to GAT and GCN.

**Strengths:**

The problem studied in this paper is significant, i.e., analyzing the sample complexity of selecting a proper hyperparameter for graph-based semi-supervised learning algorithms, since the selection of hyperparameter has a significant effect on the performance of learning algorithms. From my own perspective, the main contribution of this paper is the analysis for three classical label propagation algorithms, where the authors present both upper and lower sample complexity bounds for each of them. Particularly, the upper and lower bounds are matched, which makes the theoretic results convinced. And, the proof technique of deriving the lower bounds is novel and interesting, where the authors carefully construct a hard learning example.  I believe that these results could bring some new insights to the community.

**Weaknesses:**

- The theoretic analysis in this paper only consider single real-valued hyperparameter, e.g., the one control the trade-off between the
 local and the global consistency or the interpolation between two update rules. This significant limits the application of the theoretic results. Particularly, this kind of analysis is less sufficient for modern GNN models. There exist many hyperparameters in the optimization algorithm used for training, e.g., the learning rate and the weight decay in Adam, and they also have significant impact on the performance of model. It seems that the technique used in this paper could not be easily extend to these cases since the Rademacher complexity could not directly reflect the impact from learning algorithms. Also, I think that the title of this paper seems to exaggerate its actual workload.
- The analysis for SGC and GCAN seems to rely on the technique and assumption used in (Garg et al., 2020), i.e., treating each node as a computation tree and requiring that these trees are independent to each other. This assumption seems too strong for GNN models and raises a gap between theory and practice. Indeed, the training and evaluation of model GNNs follows the transductive learning setting [1,2], i.e., some nodes are sampled without replacement from the graph and their labels are revealed to the GNN model. Therefore, the training and test nodes are dependent. I think that using the transductive learning setting [3,4,5] to conduct the theoretic analysis could be better.

[1] Gilmer et al., Neural message passing for quantum chemistry. ICML 2017.

[2] Thomas N. Kipf and Max Welling. Semi-supervised classification with graph convolutional networks. ICLR 2017.

[3] Cong et al., On provable benefits of depth in training graph convolutional networks. NeurIPS 2021.

[4] Esser et al., Learning theory can (sometimes) explain generalisation in graph neural networks. NeurIPS 2021.

[5] Kenta Oono and Taiji Suzuki. Optimization and generalization analysis of transduction through gradient boosting and application to multi-scale graph neural networks. NeurIPS 2020.

**Questions:**

Q1: It seems that the experiments result in this paper only demonstrate the performance of the proposed GCAN model, and there are no other ones about the bounds you derived. Could you provide some experiments to verify your theoretic results ?

Q2: I am concerned about some steps in the proof of Lemma C.1. In line 1004-1010, it seems that you have used the following inequalies
$$
\frac{1}{\sqrt{(d_i+\beta)(d_j+\beta)}} = \frac{1}{\sqrt{d_i d_j + \beta (d_i + d_j) + \beta^2}} \leq \frac{1}{\sqrt{C^2_{dl} + 2\beta C_{dl} + \beta^2}} = \frac{1}{C_{dl}+\beta},
$$
where the inequality comes from $d_i, d_j \leq C_{dl}$. However, the following could not be true
$$
\left\\Vert \frac{1}{\\sqrt{(d_i+\beta)(d_j+\\beta)}} - \\frac{1}{\sqrt{(d_i+\beta')(d_j+\beta')}} \right\Vert \\leq \left\\Vert \\frac{1}{C_{dl}+\\beta} - \frac{1}{C_{dl}+\beta'} \right\\Vert
$$
since the sign of the second term is negative. I think that the right one should be derived as follows:
\begin{equation}
\begin{aligned}
& \left\\Vert \frac{1}{\\sqrt{(d_i+\beta)(d_j+\\beta)}} - \\frac{1}{\sqrt{(d_i+\beta')(d_j+\beta')}} \right\Vert  \\\\
= & \left\Vert \frac{(d_i+\beta)(d_j+\\beta) - (d_i+\beta')(d_j+\\beta')}{\\sqrt{(d_i+\beta)(d_j+\\beta)}\\sqrt{(d_i+\beta')(d_j+\\beta')}[\\sqrt{(d_i+\beta)(d_j+\\beta)}+\\sqrt{(d_i+\beta')(d_j+\\beta')}]} \right\Vert \\\\
= & \left\Vert \frac{(d_i+d_j+\\beta+\\beta')(\\beta-\\beta')}{\\sqrt{(d_i+\beta)(d_j+\\beta)}\\sqrt{(d_i+\beta')(d_j+\\beta')}[\\sqrt{(d_i+\beta)(d_j+\\beta)}+\\sqrt{(d_i+\beta')(d_j+\\beta')}]} \right\Vert \\\\
\leq & \left\Vert \frac{2(C_{dh}+C_{dl})}{(\beta+C_{dl})(\beta'+C_{dl})[(\beta+C_{dl})+(\beta'+C_{dl})]} (\beta - \beta') \right\Vert \\\\
\leq & \\frac{C_{dh}+C_{dl}}{C^3_{dl}}\Vert \beta - \beta' \Vert.
\end{aligned}
\end{equation}
And other results should be revised accordingly.

---

> ### Author Response · Authors · 2024-11-21
>
> We would like to thank the reviewer for carefully reading our submission and providing insightful comments. Please find our responses below.
>
> **Other Hyperparameters:** Thank you for raising this question. In our work, we only consider the model hyperparameter, such as weights for regularization terms and architecture hyperparameter in GNN-based models. The hyperparameters used in learning and optimization, such as learning rate and weight decay, are outside the scope of this work. We have modified the abstract and introduction of our paper to clarify this.
>
> **Assumption for GNN-based Models:** Our assumption is given in the description of Proposition 4.1. Here, the independence assumption is made for the problem instance (i.e. the original $m$ problem instances, each with $n$ nodes), not the computation trees.
> Moreover, the assumption that the computation trees follow the distribution defined in line 382 is simply an analytical technique, saying that the mean of loss of these computation trees is the same as the mean of loss of the whole graph.
>
> Our work indeed adopts the transductive learning setting within each graph, where labeled and unlabeled nodes are interdependent due to their shared structure. However, when it comes to evaluating the generalizability of the learned hyperparameter, we transition to an inductive setting. Specifically, we learn the hyperparameter from $m$ graphs, each containing labeled and unlabeled nodes, and then investigate how well this hyperparameter performs on entirely new graphs.
>
> **Additional Experiment for Theoretical Results:**
> Thank you for your suggestion. In response, we have added a new set of experiments (in Section 5.1) to validate the theoretical results in Section 3.
> For the GNN families, however, directly verifying our theoretical results empirically is challenging. The derived bounds involve multiple constants that are difficult to compute explicitly, making direct validation infeasible at this stage. However, we want to emphasize that we already included the complete proofs of all theoretical results in the file, ensuring clarity and transparency.
>
> **Mistake in Proof:** Thank you for pointing it out. We have revised the proof accordingly. However, this update does not affect the result of Theorem 4.2 presented in the main body.
>
> ---
> We hope we have addressed most of your concerns. If you have any questions, please don't hesitate to reach out to us.

---

> > ### Comment · Reviewer_WNgy · 2024-11-27
> > **Reply to the Authors**
> >
> > Thanks the authors for their responses.  My concerns are mostly addressed and I have increased my score to 6.

---

### Official Review · Reviewer_32WB · 2024-10-31

**Soundness:** 3
**Presentation:** 3
**Contribution:** 3
**Rating:** 6
**Confidence:** 3

**Summary:**

This paper takes a theoretical examination of hyperparameter tuning for graph-based semi-supervised learning (GSSL) algorithms, focusing on label propagation methods and graph neural networks (GNNs). New pseudo-dimension upper bounds and matching lower bounds for hyper-parameter tuning are proved, and the Rademacher complexity bound for tuning the weight of SGC is provided, together a new model, GCAN, that interpolates between GCNs and GATs.

**Strengths:**

1. The paper offers a rigorous theoretical analysis with novel insights into hyperparameter tuning complexities. To the best of the reviewer’s knowledge, this is the first to provide generalization guarantees to the problem of hyperparameter selection.

2. It presents a unified approach to analyze different GSSL algorithms, which is commendable. The introduction of the GCAN model is innovative and empirical results support its potential effectiveness.

**Weaknesses:**

1. It is unclear about the practical usefulness of the theoretical studies. This paper only considers tuning the single real-valued hyperparameter; it is unclear whether or not the proposed theoretical guarantees and models are able to apply to learning multiple hyper-parameters.

2. The analysis is specific to certain algorithms (i.e., SGC), and it is unclear how these findings generalize to other models.

**Questions:**

1. How do the theoretical bounds scale with different dataset sizes and graph structures?

2. Can the GCAN approach be extended to interpolate between other GNN architectures?

---

> ### Author Response · Authors · 2024-11-21
>
> We would like to thank the reviewer for their time and insightful comments. Please find our responses below.
>
> **Multiple Hyperparameters:** Exploring the tuning of multiple hyperparameters is indeed an important direction for future work. As mentioned in the conclusion section (Lines 537–539), this is a topic we recognize as an intriguing area for further research. While tuning multiple hyperparameters introduces additional complexity, we anticipate that our proposed techniques could still be valuable.
> For GNN-based algorithms, we believe that Rademacher Complexity could still serve as an appropriate complexity measure. Following a similar strategy, we would aim to bound the change in predicted values as hyperparameter values vary and then apply a covering argument. For label propagation-based methods, our approach would focus on analyzing changes in predicted classes by examining the scoring matrix.
> This lays a foundation for extending our techniques to address the challenges of multi-hyperparameter tuning in various algorithmic contexts, although they are outside the scope of this work.
>
> **Generalize to Other Algorithms:** We used unified proof techniques for all three label propagation-based methods, and used a unified proof outline for both SGC and GCAN. Therefore, we believe our proof strategy could be useful when investigating other label propagation-based and GNN-based methods.
>
> **Scale with Different Dataset Size and Graph Structure:** The dependency on the graph size and graph structure are already indicated in the theoretical results. The graph size is denoted by $n$, the number of labeled and unlabeled nodes. The graph structure is indicated by the degree matrix $D$ and adjacency matrix $W$. In the bounds, they are captured by the lower and upper bounds of the degree matrix, $C_{dl}$ and $C_{dh}$.
>
> **Interpolating Other GNN Architecture:** As long as the two architectures share the same structure in part of their calculation (e.g. the update equations of GAT and GCN both have the form of activation and a summation over the feature of all neighboring vertices in the graph), we can introduce a new variable to interpolate between the two.
>
> ---
> We hope we have addressed most of your concerns. If you have any questions, please don't hesitate to reach out to us.

---

### Official Review · Reviewer_RBYk · 2024-11-02

**Soundness:** 2
**Presentation:** 2
**Contribution:** 1
**Rating:** 3
**Confidence:** 4

**Summary:**

This paper studies hyperparameter tuning in GNN frameworks through Rademacher complexity analysis.

**Strengths:**

The theoretical approach to studying hyperparameter tuning is interesting.

**Weaknesses:**

- The core Rademacher complexity analysis largely builds upon work from [1]
- The paper's positioning is unclear - whether it aims to propose a new framework competing with GAT and GCN, or purely offers theoretical analysis of hyperparameter tuning

- Results in Table 1 show minimal practical significance:

  - Many means are identical with only slight interval differences
  - Where differences exist, they appear negligible

- The definition of $n$ is inconsistent between lines 140-145 (instances per problem) and line 295 (total labeled/unlabeled datapoints)
- The independence of bounds from $n$ in Theorems 4.2 and 4.3 requires explanation
- The rationale for GCAN's additional hyperparameter search isn't justified given the marginal improvements

**Questions:**

**More Questions:**

- Does this analysis extend to node classification or graph classification in GNNs?
- Can this approach generalize to other hyperparameter tuning scenarios beyond GNNs?
- Please clarify how CIFAR-10 is treated as a graph dataset
- Could you elaborate on your work's relationship to [2]?
- What justifies the additional complexity of GCAN's hyperparameter search?

---

**References:**

- [1]: Vikas Garg, Stefanie Jegelka, and Tommi Jaakkola. Generalization and representational limits of graph neural networks. In Hal Daumé III and Aarti Singh (eds.), Proceedings of the 37th In- ternational Conference on Machine Learning, volume 119 of Proceedings of Machine Learning Research, pp. 3419–3430. PMLR, 13–18 Jul 2020.

- [2]: Hsu, Kelvin, Richard Nock, and Fabio Ramos. "Hyperparameter learning for conditional kernel mean embeddings with rademacher complexity bounds." Machine Learning and Knowledge Discovery in Databases: European Conference, ECML PKDD 2018, Dublin, Ireland, September 10–14, 2018, Proceedings, Part II 18. Springer International Publishing, 2019.

---

> ### Author Response · Authors · 2024-11-21
>
> We thank the reviewer for carefully reading our submission and providing many insightful suggestions. We are happy to address ALL your comments as below:
>
> **Rademacher Complexity proof:** We would like to highlight that the problem setting in our work is fundamentally different from that of Garg et al. Their study focuses on *graph classification* with *fixed hyperparameters*, while our work addresses *node classification* with the simultaneous learning of *both model parameters and hyperparameters*. To the best of our knowledge, our work is the first to provide a theoretical guarantee for hyperparameter tuning in GNNs. We have discussed this distinction in Lines 361-365 of the paper.
> While the proofs share a similar underlying approach, the inclusion of hyperparameters—i.e. the self-loop weight $\beta$ in SGC and the interpolation parameter $\eta$ in GCAN—adds substantial complexity to the problem we tackle.
>
> **Paper's Position:** The primary focus of our work is the *theoretical analysis* of hyperparameter selection. This is clearly indicated by our title. Also, we mentioned in line 444 that the proposed interpolation architecture GCAN is not our main contribution.
>
> **GCAN Experiments:** The purpose of GCAN is not to outperform both GCN and GAT significantly. GCAN is an initial step towards solving the more complex problem of automatically selecting from multiple model architecture. It is not surprising that, when GAT and GCN perform similarly, the improvement of GCAN with optimal $\eta$ does not differ from that of other $\eta$ values by much.
> Additionally, we want to emphasize that, regardless of the practicability of GCAN, the theoretical guarantees provided in our Theorem 4.1 and 4.2 are themselves novel.
>
> **Definition of $n$:** $n$ refers to the "total labeled and unlabeled nodes", which is equivalent to the "number of instances per problem". Note that $n$ is different from $m$. $m$ is defined as the total number of problem instances (graphs), where each problem instance contains $n$ nodes.
>
> **Dependence on $n$ in Theorem 4.2, 4.3:** There are no direct dependences on $n$, but the dependence on the number of nodes is implicitly captured by the more fine-grained value $C_{dl}, C_{dh}$, and $C_Z$. Here, $C_{dl}$ and $C_{dh}$ are the lower and upper bounds of the degree of the nodes. For a larger $n$, $C_{dl}$ and $C_{dh}$ should be larger in general. $C_Z$ is the Frobenius norm of the feature matrix $Z \in \mathbb{R}^{n \times d}$. Since the size of $Z$ scales with $n$, the value of $C_Z$ is generally larger for larger $n$.
> We really appreciate this question. We have added a Remark to clarify this.
>
> **Rationale for GCAN's Hyperparameter:** In practice, neither GAT nor GCN consistently outperforms the other, as demonstrated by Dwivedi et al. (2023). To address this, we introduce the hyperparameter $\eta$, which enables the automatic selection between GAT and GCN, effectively combining their strengths. This motivation was included at the beginning of Section 4.2.
>
> Vijay Prakash Dwivedi, Chaitanya K Joshi, Anh Tuan Luu, Thomas Laurent, Yoshua Bengio, and Xavier Bresson. *Benchmarking graph neural networks.* Journal of Machine Learning Research, 24(43):1–48, 2023.
>
> **Extension to Node and Graph Classification:** We are indeed investigating hyperparameter selection for *node classification* in GNN. The analysis in Garg et al. is for *graph classification* with fixed hyperparameters. We agree that extending analysis on hyperparameter tuning to graph classification is an interesting direction for future research. We believe our approach could serve as a valuable foundation or inspiration for tackling such problems.
>
> **Methods Beyond GNN:** We would like to highlight that we have already explored *label propagation-based methods* in section 3 of our paper, which is *fundamentally different from GNN*. Label propagation-based algorithms and GNN-based algorithms are two major graph-based semi-supervised learning methods. We believe that the insights and arguments presented in our work are applicable and valuable for any algorithms of these two types as well as other related methods.
>
> **CIFAR10 as Graph Dataset:** For CIFAR10, we first calculate the Euclidean distance between the feature vectors of each pair of nodes and then put an edge between the two nodes if their distance is smaller than a threshold. We apologize for this confusion. We have added this clarification to the description of the experiment setup in Appendix D.

---

> > ### Author Response · Authors · 2024-11-21
> >
> > **Relationship to [2]:** Hsu et al. studies kernel hyperparameter for the problem of Conditional Kernel Mean Embeddings. Their problem setting is closer to the setting in Balcan and Sharma (2021), where the graph is not fixed and they tune "Graph kernel" hyperparameters. Our work focuses on the algorithm hyperparameters instead, i.e. regularization hyperparameters in label propagation-based methods and architectural hyperparameters for GNN-based methods, and considers the graph is fixed and given to us.
> >
> > Maria-Florina Balcan and Dravyansh Sharma. *Data driven semi-supervised learning.* Advances in Neural Information Processing Systems, 34, 2021.
> >
> > **Additional Complexity of GCAN:** As explained in our Remark 2, GCAN has a larger complexity than SGC because the model is more complicated. Specifically, the parameter dimension of SGC is smaller than GCAN, which leads to the dependency on feature dimension $d$ being smaller. However, this added complexity is not a drawback. Instead, it enables GCAN to handle more challenging problem instances effectively, often leading to improved performance.
> >
> > ---
> > We hope we have addressed most of your concerns. If you have any questions, please don't hesitate to reach out to us.

---

### Official Review · Reviewer_oxKk · 2024-11-04

**Soundness:** 3
**Presentation:** 3
**Contribution:** 3
**Rating:** 6
**Confidence:** 3

**Summary:**

This paper delves into the theoretical analysis of hyperparameter selection in graph-based semi-supervised learning, specifically focusing on label propagation algorithms and modern graph neural networks.

**Strengths:**

- The paper presents a novel theoretical analysis of hyperparameter selection in graph-based semi-supervised learning, particularly for GNNs.
- The theoretical guarantees can guide the design of efficient and robust hyperparameter selection methods in practice.

**Weaknesses:**

- The experiment in the main body of the paper is limited.
- Considering the proposed GCAN is part of the central contribution of this paper, the comparison between GCAN and baselines should be included in the main body instead of the appendix.

**Questions:**

How do the proposed methods compare to existing hyperparameter tuning techniques, such as grid search and Bayesian optimization, in terms of efficiency and accuracy?

---

> ### Author Response · Authors · 2024-11-21
>
> We would like to thank the reviewer for their time and insightful comments. Please find our responses below.
>
> **Experiments in the main body (Weakness 1 & 2):**
> Thank you for your suggestion. We have added two sets of experiments in the main body. One experiment is to validate the bound we derived for label propagation-based algorithms, and the other is to show the effectiveness of our hyperparameter selection framework using GCAN.
> However, we do want to emphasize that the primary focus of our work is the *theoretical analysis* of hyperparameter selection, as emphasized in our title. We also noted in Line 444 that the proposed interpolation architecture, GCAN, is not the main contribution of our work.
>
> **Question:**
> Our primary contribution lies in the theoretical analysis. We provide upper and lower bounds on the sample complexity of any algorithm for tuning the hyperparameters. Therefore, we did not directly compare our method with standard approaches such as grid search or Bayesian optimization. However, we agree that testing and benchmarking hyperparameter tuning methods would be an interesting avenue for future work.
> The results in Table 1 show the prediction error of GCAN across different $\eta$ values, so it effectively captures the outcomes of a grid search approach.
>
> ---
> We hope we have addressed most of your concerns. If you have any questions, please don't hesitate to reach out to us.

---

> > ### Comment · Reviewer_oxKk · 2024-11-27
> >
> > Thanks for your response. Based on the replies, I would keep my positive rating.

---

### Author Response · Authors · 2024-11-21
**Revisions to the Manuscript**

We sincerely thank all reviewers for their careful evaluations and constructive feedback. Below, we summarize the updates made to our manuscript based on your suggestions. The changes are colored blue.
1. Clarifications in Abstract and Introduction: We updated these sections to clarify that our work focuses exclusively on model or architecture hyperparameters, excluding learning hyperparameters such as the learning rate.
2. Expanded Experiment Section: We added new experiments to strengthen our contributions.
    1) Label Propagation-Based Family (Normalized Adjacency Matrix-based Family): To empirically verify the theoretical bounds presented.
    2) GNN Family (GCAN): To demonstrate that selecting the hyperparameter $\eta$ via backpropagation performs no worse than GAT or GCN, and can be used to automatically select between the two architectures.
3. Proof Adjustments in Appendix: We corrected details in the proofs provided in the appendix. These adjustments do not affect the main results presented in the manuscript.
4. Refinements for Conciseness: Certain texts were rephrased to ensure the manuscript adheres to the 10-page limit without compromising clarity or content.

Thank you again for your valuable input. We look forward to hearing your thoughts and engaging in further discussions!

---

### Author Response · Authors · 2024-11-26

Dear Reviewers,

Thank you once again for your thoughtful comments and valuable suggestions! As the deadline for the PDF revision approaches, we kindly ask if you could let us know whether our previous responses have sufficiently addressed your concerns.

We are always more than happy to provide further clarifications or address any additional questions you may have. We greatly value your feedback and look forward to your insights.

Best regards,

Authors

---

### Meta-Review · Area_Chair_iiPD · 2024-12-18

**Metareview:**

This work explores the problem of tuning hyperparameters in graph-based semi-supervised learning algorithms, including classical label propagation and modern graph neural networks. The authors derive novel upper and matching lower bounds for the pseudo-dimension of hyperparameter selection in label propagation algorithms, providing insights into the data requirements for learning effective parameters. Additionally, they extend this study to graph neural networks, offering Rademacher complexity bounds for tuning self-loop weights in Simplified Graph Convolution (SGC) networks and propose a tunable architecture that interpolates between Graph Convolutional Networks (GCN) and Graph Attention Networks (GAT).

The paper addresses an important problem, and the theoretical results are interesting and useful.  However, its practical impact is still uncertain.  For example, only one hyperparameter is considered, and the experiments are quite limited in scale and baselines.  The reviewers raised a few technical questions that the rebuttal has quite well addressed.  Overall, another round of review appears appropriate for this paper.

**Additional Comments On Reviewer Discussion:**

The rebuttal has been noted by the reviewers and have been taken into account by the AC in the recommendation of acceptance/rejection.

---

### Decision · Program_Chairs · 2025-01-22

Reject